# Southern Ocean cloud and shortwave radiation biases in a nudged climate model simulation: does the model ever get it right?

Sonya L. Fiddes[1,2], Alain Protat[3,1], Marc D. Mallet[1], Simon P. Alexander[4,1], and Matthew T. Woodhouse[2,1]

[1]Australian Antarctic Program Partnership, Institute for Marine and Antarctic Studies, University of Tasmania, Hobart, Australia
[2]Climate Science Centre, Oceans and Atmosphere, Commonwealth Scientific and Industrial Research Organisation, Aspendale, Australia
[3]Bureau of Meteorology, Melbourne, Australia
[4]Australian Antarctic Division, Hobart, Australia

**Correspondence:** Sonya Fiddes (sonya.fiddes@utas.edu.au)

**Abstract.**

The Southern Ocean radiative bias continues to impact climate and weather models, including the Australian Community Climate and Earth System Simulator (ACCESS). The radiative bias, characterised by too much shortwave radiation reaching the surface, is attributed to the incorrect simulation of cloud properties, including frequency and phase. To identify cloud regimes important to the Southern Ocean, we use $k$-means cloud histogram clustering, applied to a satellite product and then fitted to nudged simulations of the latest generation ACCESS atmosphere model. We identify instances when the model correctly or incorrectly simulates the same cloud type as the satellite product for any point in time or space. We then evaluate the cloud and radiation biases in these instances.

We find that when the ACCESS model correctly simulates the cloud type, cloud property and radiation biases of equivalent, or in some cases greater, magnitude remain compared to when cloud types are incorrectly simulated. Furthermore, we find that even when radiative biases appear small on average, cloud property biases, such as liquid or ice water paths or cloud fractions remain large. Our results suggest that simply getting the right cloud type (or the cloud macrophysics) is not enough to reduce the Southern Ocean radiative bias. Furthermore, in instances where the radiative bias is small, it may be so for the wrong reasons. Considerable effort is still required to improve cloud microphysics, with a particular focus on cloud phase.

## 15  Key Figure

(used for advertising the paper)

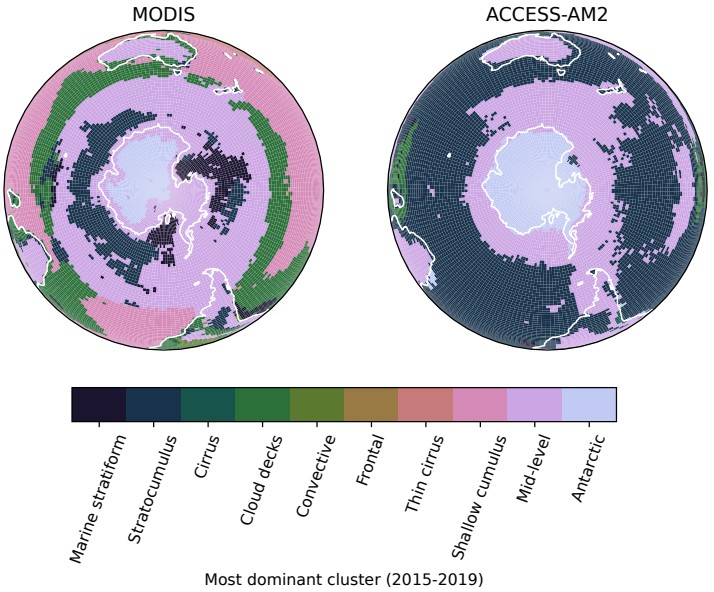

**Figure 0.** The most dominant cloud regime for each gridbox for the MODIS product (left) and the ACCESS-AM2 model (right)

# 1 Introduction

Global climate models, including coupled, Earth system and atmosphere-only models, have presented a significant radiative bias over the Southern Ocean (SO) for a number of generations, as documented by the Coupled Model Intercomparison Project (CMIP) phases 3, 5 and 6 (Bodas-Salcedo et al., 2014; Schuddeboom and McDonald, 2021). This bias has also been found in numerical weather prediction (NWP) models (Protat et al., 2017; McFarquhar et al., 2021). The radiative bias largely been found to be a result of poor simulation of cloud properties within models, in particular, the cloud phase in the cold sector of extra-tropical cyclones (Haynes et al., 2011; Bodas-Salcedo et al., 2016b). Over the SO, clouds are ubiquitously made up of supercooled liquid water droplets, setting them apart from clouds on other parts of the Earth (Huang et al., 2012; Chubb et al., 2013; Mace and Protat, 2018; Listowski et al., 2019). This distinction creates a key problem for the simulation of SO clouds given that the majority of cloud and aerosol microphysical parameterisations have been developed from observations collected in regions of the world where supercooled liquid water droplets do not occur frequently. The resulting positive surface radiative bias caused by the poor simulation of cloud characteristics has implications for numerous aspects of the global and regional climate system, including on the global energy budget (Haynes et al., 2011; Frey et al., 2017; Schuddeboom and McDonald, 2021), the strength and positioning of the Intertropical Convergence Zones (ITCZ) (eg. Hwang and Frierson (2013), debated in Hawcroft et al. (2017)) and the jet streams (Kay et al., 2016; Hawcroft et al., 2017) as well as SO sea surface temperatures (SSTs) (Hyder et al., 2018).

Within the cloud-radiation bias literature, model evaluation techniques now commonly involve both the use of an observational simulator (satellite or ground based) to ensure that model and observed products are comparable (eg. Bodas-Salcedo et al., 2011; Kuma et al., 2020), and a method to characterise cloud types and/or the associated synoptic conditions (eg. Williams and Webb, 2009; Field and Wood, 2007). Many of these cloud characterisation techniques have focused on a clustering-based approach using cloud histogram products from the Cloud Feedback Model Intercomparison Project (CFMIP) Observation Simulator Package (COSP), utilising various satellites and models. Examples of both $k$-means clustering (Jakob and Tselioudis, 2003; Williams and Tselioudis, 2007; Williams and Webb, 2009; Tselioudis et al., 2013; Oreopoulos et al., 2014; Mason et al., 2015; Oreopoulos et al., 2016; Leinonen et al., 2016; Tselioudis et al., 2021; Cho et al., 2021) and self-organising maps (SOMs) (McDonald et al., 2016; Schuddeboom et al., 2018) are found in the literature, often presenting comparable cloud regime structures.

For example, Tselioudis et al. (2021), in analysing the CMIP6 model suite, find an improvement from CMIP5 in simulating global mean 'weather state' distributions (i.e. cloud regimes derived via k-means clustering from the updated International Satellite Cloud Climatology Project - ISCCP - cloud histograms). Shallow-cumulus clouds are found to be consistently underestimated by a selection of CMIP6 models, while for the other weather states described, a larger model spread is found (Tselioudis et al., 2021). However, they note, as does Schuddeboom et al. (2018), that averaging masks important regional biases, such as those over the SO. Schuddeboom and McDonald (2021) in fact find important differences in cluster frequency biases for the SO compared to the global values in CMIP6 models. Of particular interest, they find that stratocumulus clouds are occurring too frequently over the SO in the CMIP6 models, but compensate for the associated positive radiative bias by

not being bright enough, which is opposite to CMIP5 findings of too few and too bright clouds. Schuddeboom and McDonald (2021) also note that the models with the largest compensating errors tend to have the lowest radiation biases over the SO, indicating that updated parameterisations and tuning is causing the 'right' answer for the wrong reasons. This finding is particularly concerning, given the effort to reduce the SO radiative bias so far. Similar conclusions have been made by Gettelman et al. (2020). Additionally, Zelinka et al. (2020) find that CMIP6 generation models have a higher climate sensitivity, in part due to changes in how SO cloud microphysics are now treated.

Other studies have developed regimes dependant on cloud dynamic or thermodynamic properties, such as cyclone compositing techniques (Field and Wood, 2007) or mid-tropospheric large-scale vertical motion (Bony and Dufresne, 2005). Importantly, for many subsequent studies, the evaluation methods described in such papers as listed above are used to understand biases beyond that of the cloud histograms, (eg. radiative biases or super-cooled liquid water biases). For example, the Bodas-Salcedo et al. (2014) work relies on the methods presented by both Williams and Webb (2009) and Field and Wood (2007).

The majority of the aforementioned cloud-regime studies have compared free-running simulations, such as those performed for the CMIP experiments, including the Atmospheric Model Intercomparison Project (AMIP) where sea surface temperatures and sea ice concentrations are prescribed. However, with free-running simulations, the depth of analysis is limited as the synoptic scale meteorology cannot be considered the same. Some studies have used synoptic compositing to alleviate this issue, where certain synoptic situations can be compared like-for-like, and location and timing is then considered irrelevant. However, these studies are often limited to one synoptic type, such as cyclone centers, ignoring a number of other synoptic situations relevant for the SO, as well as any compensating errors that may exist. Additionally, focusing on just one synoptic condition follows a pre-conceived idea of the error, which may or may not hold true for newer model generations. Alternatively, many studies use a decomposition of the radiative bias into three components: a term that quantifies the bias in the frequency of occurrence of cloud clusters, a term for the bias in the radiative balance itself, and a second order co-variation (Williams and Tselioudis, 2007; Williams and Webb, 2009; Mason et al., 2015; Schuddeboom et al., 2018; Schuddeboom and McDonald, 2021). This decomposition only considers the overall means or frequencies of occurrence, and so day-for-day comparison is not required. With this decomposition, both Mason et al. (2015) and Schuddeboom et al. (2018) found that the largest proportion of the radiative bias was explainable by biases in the cloud frequency of occurrence. This has led Schuddeboom et al. (2018) to further speculate that clouds themselves may be 'simulated well', but their distributions are wrong, leading to large errors in radiative biases. However this has not been explored further.

In this paper, we present an in-depth evaluation of the SO radiative bias of the Australian Community Climate and Earth System Simulator (ACCESS) - Atmosphere-only Model Version 2 (AM2). Importantly, we run a nudged simulation, so the model can be compared to satellite products directly in time and space. We then use cloud regime clustering to understand the underlying reasons for the observed cloud and radiative biases. By using a nudged simulation, we are able to composite and evaluate days and locations when cloud regimes are correctly identified by the model (ie. are the same as what was seen by the satellite), as well as instances when the model incorrectly simulates the cloud regime. We aim to answer the following question: if the model simulates the correct cloud structure for the time and place, is the radiative bias improved? This work provides the essential first step towards a long-term goal of improving the ACCESS model (and possibly the Unified Model

family) and the representation of clouds over the SO. The methods outlined in this work will be used to inform and further evaluate any developments made to the model to ensure that any resulting changes in the model are understood.

## 2 Data and methods

### 2.1 ACCESS-AM2 model setup

The ACCESS-AM2 model is an atmosphere-only configuration of the ACCESS-CM2 coupled climate model and is described in Bodman et al. (2020). The atmospheric component of ACCESS is the Unified Model (UM) at vn10.6, GA7.1, which is fully described in Walters et al. (2019). In brief, the radiation scheme used in ACCESS-AM2 is the Suite Of Community RAdiative Transfer codes based on Edwards and Slingo (SOCRATES) (Edwards and Slingo, 1996), and is called hourly. The prognostic cloud fraction and condensate (PC2) cloud scheme is used (Wilson et al., 2008), which includes large-scale and convective clouds. The convective scheme, including downdraughts and momentum transport, is based on Gregory and Rowntree (1990). Further details on ACCESS-CM2 and GA7.1 configurations can be found in Bi et al. (2020) and Walters et al. (2019). In the atmosphere-only configuration, we prescribe sea surface temperature (SST) and sea ice concentration (SIC) following the CMIP6 AMIP protocol (Eyring et al., 2016). The input fields can be found at the input4MIPs webpage (https://esgf-node.llnl.gov/projects/input4mips/) and the data method is described in Hurrell et al. (2008). Solar forcing, greenhouse gases, volcanic aerosol optical depth, aerosol chemistry emissions and ozone are also prescribed according to CMIP6 (Eyring et al., 2016) and available at the input4MIPs site.

ACCESS-AM2 includes the GLOMAP-mode (GLObal Model of Aerosol Processes) aerosol microphysical scheme (Mann et al., 2010, 2012), including parameterised sulfur chemistry driven by prescribed oxidants. GLOMAP-mode is a two-moment, pseudo-modal microphysical aerosol scheme, representing four soluble (nucleation, Aitken, accumulation and coarse) and one insoluble (Aitken) aerosol modes. Each mode is internally mixed. GLOMAP uses a scaled ($1.7\times$, as recommended in Mulcahy et al., 2018) DMS surface water monthly climatology (Lana et al., 2011) combined with the Liss and Merlivat (1986) flux parameterisation (also see Fiddes et al., 2018). Sea salt emissions are calculated online and occur following the Gong (2003) parameterisation. Dust is treated separately according to Woodward (2001), using a six bin scheme.

Historical (pre-2014) anthropogenic aerosol emissions are provided by the Community Emissions Data System (Hoesly et al., 2018) and biomass burning by the Global Fire Emissions Database with small fires (GFED4s) (Van Marle et al., 2017). Post 2014, the shared socioeconomic pathway (SSP) 2-4.5, a 'middle of the road emissions pathway' (Fricko et al., 2017) is used, with emissions developed by the Integrated Assessment Models Consortium and described in Feng et al. (2020).

The land-surface scheme in ACCESS-AM2 is the Community Atmosphere Biosphere Land Exchange (CABLE) version 2.5 land surface model (also described in Bi et al., 2020). ACCESS-AM2 runs at a 1.25x1.875 degree horizontal resolution with 85 vertical levels and uses the 'ENDGame' dynamical core (Wood et al., 2014). Further information on the general model setup and preliminary model evaluation against standard climate fields, such as surface air temperatures, sea surface temperature, rainfall, mean sea level pressure (MSLP) and precipitation can be found in Bi et al. (2020) and Bodman et al. (2020).

The simulation presented in this study has been nudged to the European Centre for Medium-range Weather Forecasting (ECMWF) Reanalysis 5 (ERA5) product (Hersbach et al., 2020). Nudging occurs at every dynamical time step from reanalysis fields that are updated every three hours for the horizontal winds and temperature in the free troposphere and stratosphere. In this study, output is created at monthly and daily mean resolution for the years 2015-2019 (chosen to overlap with recent Southern Ocean field campaigns described in McFarquhar et al. (2021)).

## 2.2 Observational products

Simulated cloud fields are evaluated against the Moderate Resolution Imaging Spectroradiometer (MODIS) Combined Aqua/Terra, Level 3 daily, 1x1 degree grid, Collection 6.1, COSP product (MCD06COSP_D3_MODIS) (Platnick et al., 2017). This product has been derived specifically for use in model evaluation using the COSP outputs for CMIP6 and is described in Hubanks et al. (2020). Available properties include cloud optical depth ($\tau$) for total, ice and liquid clouds, cloud top pressure (CTP), cloud mask fraction (derived from pixel-level cloudiness assessments) for total, low, mid and high clouds, cloud retrieval fraction (derived from the successful retrieval of $\tau$) for total, ice and liquid clouds, cloud effective particle radius ($R_{eff}$) for ice and liquid clouds, liquid water path (LWP), ice water path (IWP) and the joint histogram for CTP and $\tau$ for cloudy and partly cloud pixels. All cloud properties in the MCD06COSP_D3_MODIS product are for day-time only scenes. Full descriptions of these cloud properties can be found in Pincus et al. (2012); Platnick et al. (2017) and Hubanks et al. (2020).

Collection 6 now includes partly cloudy scenes that represent heterogeneous broken cloudy or cloud edge pixels (Platnick et al., 2017). Successfully retrieved cloudy pixels are considered to be high quality, where as partly cloudy scenes have a higher rate of retrieval failure of 34%, and are slightly less robust (Platnick et al., 2017). Collection 6 has shown a marked improvement upon Collection 5 in part due to the re-writing of the cloud optical property retrievals, resulting in an increase in cloud phase classification of 10%, and a 90% agreement in total cloud phases between MODIS and the CALIOP (Cloud Aerosol Lidar With Orthogonal Polarization) retrievals for cloudy pixels (Platnick et al., 2017; Marchant et al., 2016). The largest improvement in cloud phase detection was found for opaque clouds over ice or snow covered areas, whilst detection for thin cloud retrievals over warm or bright surfaces remain an issue. Improved optical retrievals have also reduced the biases in the $R_{eff}$, however, it has been noted that much of the evaluation performed has been for single layered clouds, with multi-layered clouds remaining un-assessed (Marchant et al., 2016).

Comparisons of MODIS retrievals to aircraft field campaigns over the north-east and south-east Pacific, as well as ground-based observations in Finland have indicated that liquid $R_{eff}$ is overestimated, which impacts the LWP retrievals (Painemal and Zuidema, 2011; King et al., 2003; Sporre et al., 2016; Noble and Hudson, 2015; Min et al., 2012). However, a number of studies also noted that there is good agreement in variability between liquid $R_{eff}$ and LWP (Min et al., 2012; Noble and Hudson, 2015). The cloud retrieval fraction and cloud optical depth has been found to perform better and be strongly correlated with field observations (Sporre et al., 2016; Noble and Hudson, 2015). Ice cloud retrievals have been known to be more difficult for a number or reasons relating to crystal shape and scattering properties as well as lifetime and expanse. Evaluation of the MODIS Collection 5 ice retrievals found an overestimation compared to an infrared radiative closure method for determining ice $\tau$ (Holz et al., 2016). The updates to the optical retrieval method in Collection 6 has reduced the bias to a more satisfactory

level. We note that the evaluations discussed above have not been performed for the specific COSP product we are using in this work, but give us an idea of the MODIS satellite retrievals overall performance.

In this work, we are using cloud retrieval fraction, which we subsequently refer to as cloud fraction (CF), for ice (CFI) and liquid clouds (CFL) and the joint histogram for CTP and $\tau$ for cloudy and partly cloudy pixels. Following the methods of Oreopoulos et al. (2016); Schuddeboom et al. (2018) and Saponaro et al. (2020) we use the combined joint histogram product (i.e. the sum of the cloudy and partly cloud products). Very large biases that we considered unrealistic were found for the modelled $R_{eff}$ and hence these fields have not been used for this work. Similar biases were found for the LWP and IWP, and hence, the COSP-derived products (described in the next section) for these fields were not used, but replaced with the raw model output. While this adds a degree of uncertainty to this work, we believe such an analysis with the derived COSP fields would not have been useful.

Furthermore, care has been taken to ensure that we are comparing only grid-box mean values of the LWP and IWP. Model outputs (both COSP and raw) are provided directly as grid-box mean, while the MODIS products are provided as in-cloud mean values. We have performed the appropriate conversions where grid-box mean is equal to the in-cloud mean multiplied by cloud fraction. After comparison of both grid-box mean and in-cloud mean values, we have chosen to use the grid-box mean values. This choice does effect some of the results of this study, as would choosing to use the in-cloud values instead. However, we believe this choice is robust for two reasons: firstly, grid-box means are the native model output and this is a model evaluation study and secondly, the grid-box means showed a better model performance than the in-cloud mean values, likely due to the weighting of the cloud fraction.

For this analysis we have removed all instances of clear sky from both the model and the observed histograms. We have done this by finding the instances where the CTP-$\tau$ histograms summed to zero. While this allows us to focus on instances when the model is simulating cloud, either correctly or incorrectly, it also means we are not considering instances when either the model or satellite simulated cloud while the other simulates clear sky. In this work, we are considering grid boxes of 1.25x1.875 degrees and daily means, meaning that there are very few instances of clear sky occurring for a full day over a large domain. On average, there are less than 1.5% of grid boxes simulated as clear sky in ACCESS-AM2 over the Southern Ocean, and far fewer in the satellite (in part due to the addition of the partly cloudy optical depth retrievals as well). For these reasons, we believe this choice is robust.

To evaluate the radiative bias in ACCESS-AM2, we use the Clouds and the Earth's Radiant Energy System (CERES) Syn1Deg product (Doelling et al., 2013, 2016) for evaluation at daily timescales. The bias in the outgoing top of atmosphere (TOA) shortwave (SW) cloud radiative effect (CRE) (SWCRE$_{TOA}$) has been chosen for analysis in this work based on previous findings from the literature presenting the SWCRE as the most problematic aspect of the SO energy balance. The CRE presented in this work is the difference between the clear-sky radiation and the all-sky radiation fields (for both the model and satellite products). Throughout this paper, a predominantly positive SWCRE$_{TOA}$ is present, e.g. the ACCESS-AM2 model is allowing too much shortwave radiation to pass through the clouds and not reflecting enough shortwave radiation out to space via clouds, corresponding with too much short wave radiation reaching the surface.

## 2.3 COSP

To directly compare the ACCESS-AM2 cloud properties to that of satellite products, we use the COSP, described by (Bodas-Salcedo et al., 2011), as prescribed for simulations within the CFMIP activity of CMIP6 (Webb et al., 2017). The MODIS products provided the best coverage for the time period of interest to this study. For this reason, no other satellite products are considered in this work. Simulated fields include the joint histograms of CTP and $\tau$, as well as liquid and ice cloud fractions, water paths and $\tau$. Using COSP output allows the appropriate comparison of model to satellite products. This method applies the assumptions and limitations of the satellite algorithms to the model output, limiting the possibility that biases are due to differences in processing. For the LWP and IWP, we are considering the grid-box mean (as opposed to the in-cloud mean). We note that in the IWP and LWP fields, significantly large and seemingly unrealistic biases between the model COSP product and the MODIS product were found. IWP and LWP are reliant on $R_{eff}$ retrievals, which as discussed above, are less well captured in satellite products. For this reason, as stated above, the actual simulated (i.e. raw model output) IWP and LWP are used for this analysis instead of the COSP derived product.

## 2.4 Cloud regime clustering

In this study, we use $k$-means clustering to derive cloud regimes (Anderberg, 1973). $K$-means clustering is a form of unsupervised machine learning that separates $N$ points in $k$ clusters by minimising the sum of squared distances within each cluster. In this case, the Euclidean distance is considered, which is equivalent to the minimisation of variance (or inertia) within each cluster. To perform this analysis, the SciKit Learn (Pedregosa et al., 2011) and Dask-Machine Learning python packages were used, where $k$-means clustering has been implemented for distributed computing.

We apply $k$-means clustering to four years (2015-2019) of daily MODIS histogram data (CTP/$\tau$), over the entire globe. We have normalised the CTP/$\tau$ histograms to one (as opposed to the cloud fraction) to limit the impact of cloud fraction biases within the ACCESS-AM2 model on the clustering results. Whilst this impacts our ability to compare to other studies, it allows the clustering to target cloud vertical extent and thickness regardless of the total fraction and how well it is captured by the ACCESS-AM2 model.

A complication of clustering methods, including $k$-means, is the choice in the number of clusters. While there is no 'right' or 'wrong' answer to the number of clusters to select, there are practical considerations and statistical metrics that can help guide this choice. Three statistical metrics were applied to this work in an attempt to aid the decision on the number of clusters to choose, including the 'Elbow' method (Wilks, 2011), the Calinski-Harabasz (CH) index (Calinski and Harabasz, 1974) and the Davies-Bouldin (DB) index (Davies and Bouldin, 1979). The SciKit Learn application programming interface provides a detailed explanation of each metric in addition to their advantages and disadvantages. Unfortunately, the statistical guidance provided by these metrics was not useful in cluster number selection (suggesting 2, 4 and 17 clusters respectively). After consideration of a range of choices, we selected 12 clusters.

We then used the MODIS cluster regimes to predict the cloud regimes of ACCESS-AM2, by fitting each ACCESS-AM2 data point to one of the 12 cluster centres. We chose this method, as opposed to a hybrid approach as taken by Mason et al.

(2015), so that we could apply the same cluster centres to multiple model simulations, allowing a direct comparison over a

number of simulations. A similar method is described in Williams and Webb (2009), where they also point out that this method eliminates further subjective choices.

We have chosen to perform clustering over the entire globe, though only results for the SO (defined for this work as the broad region from 30-69°S) are examined in this current study. This choice will allow us to assess if any changes applied to the model focused on improving Southern Ocean clouds have unintended effects outside this region. For this reason, while the

cloud-regime histograms are defined globally, only those important to the SO will be examined in detail.

## 2.5  Bias decomposition

To explore cloud property biases, we follow the decomposed bias metrics described in Williams and Webb (2009); Mason et al. (2015) and Schuddeboom et al. (2018) where the bias ($\Delta$: ACCESS-AM2 − satellite) of a simulated field of interest $F$ for each cluster $r$ can be summarised as the errors due to the RFO ($F_r^{sat} \cdot \Delta RFO_r$, referred to as RFO errors) plus the errors in the

simulated field ($\Delta F_r \cdot RFO_r^{sat}$, referred to as field errors) plus a second order co-variation term ($\Delta F_r \cdot \Delta RFO_r$) otherwise referred to as the cross-term.

$$\delta F_r = F_r^{sat} \cdot \Delta RFO_r + \Delta F_r \cdot RFO_r^{sat} + \Delta F_r \cdot \Delta RFO_r \tag{1}$$

## 3  The ACCESS-AM2 radiation and cloud biases

The annual and seasonal ACCESS-AM2 SWCRE$_{TOA}$ and cloud biases compared to the CERES-Syn1D or MODIS products

are shown in Figure 1. The boundaries of our analysis, shown by the dashed lines, represent three regions of interest: the mid-latitudes defined as 30-43°S, the sub-polar region defined as 43-58°S, and the polar region defined as 58-69°S of the SO. While seasonal expansion and contraction of the Ferrel and Polar atmospheric cells means these stationary boundaries may not capture the seasonal bias boundaries perfectly, for the purpose of this work they are satisfactory. These regions will be used throughout the rest of this work to explore how differently cloud types are contributing to the SWCRE$_{TOA}$ bias in each region.

Considering firstly the SWCRE$_{TOA}$ (row 1, Figure 1), a persistent positive bias in the polar region of the SO is found across all seasons. This bias has not been improved upon in this latest generation model, with a similar total SW$_{TOA}$ radiative bias (not shown) to that reported in a previous version of ACCESS presented in Fiddes et al. (2018). The summer (DJF), shown in Figure 1c1, continues to have the largest polar bias, while the winter (JJA) season has the smallest bias overall, in agreement with previous work (Bodas-Salcedo et al., 2012; Kuma et al., 2020). A positve/negative/positive 'tripole' can broadly be seen in

the annual mean biases (b1), where the SWCRE$_{TOA}$ is on average overestimated in the mid-latitudes, weakly underestimated in some parts of the sub-polar SO, and largely overestimated in the polar SO. MAM and JJA (Figure 1d1, e1) show very weak positive biases in the polar region, though with well defined transitions from negative to positive in the sub-polar to mid-latitude regions. In SON and DJF, the positive biases in the polar and mid-latitude regions are well defined, while the negative sub-polar biases becomes far less zonally coherent. Some strongly positive regions in spring are also found along the edge of the sea-ice

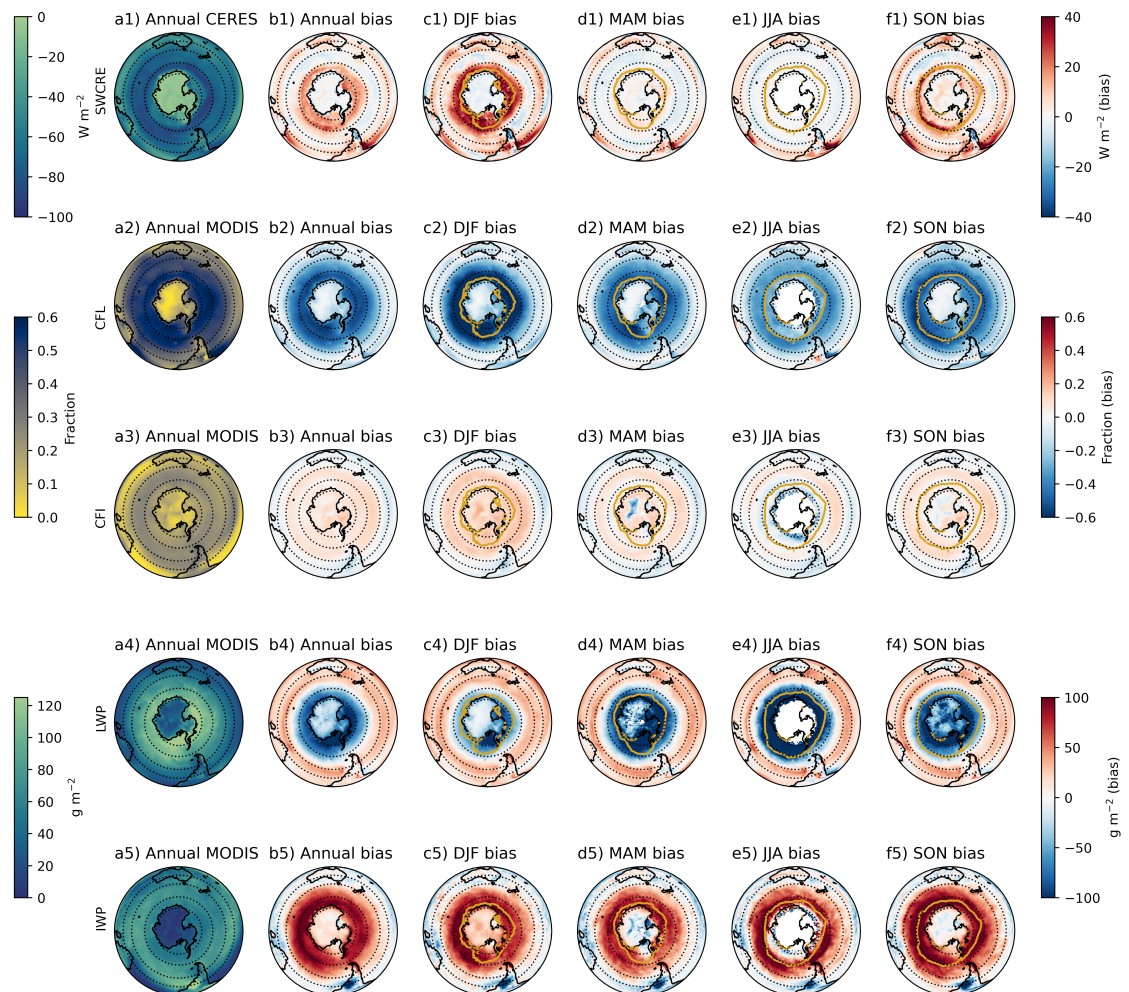

**Figure 1.** Column a) the annual mean satellite (CERES-Syn1D or MODIS) derived field for row 1: SWCRE$_{TOA}$, (W m$^{-2}$) 2: CFL 3: CFI , 4: LWP (g m$^{-2}$ ) and 5: IWP (g m$^{-2}$ ); column b) annual biases (ACCESS-AM2 - satellite) in the same respective fields; c-f) seasonal biases (DJF, JJA, MAM,SON). The yellow line in figure c-f) shows the mean seasonal extent of sea ice (calculated at 15% coverage), taken from the Hurrell et al. (2008) AMIP sea ice concentration data set. Dashed lines are located at 30°S, 43°S, 58°S, and 69°S

zone (yellow line). This feature could be due to some discrepancy between the sea-ice concentrations prescribed to the model and what was seen by the CERES satellite product, or could be issues with the satellite retrieving observations over ice-covered areas. Kuma et al. (2020) also noted a strong latitudinal dependence of radiative biases in a similar version of the UM and a reanalysis product. While much of the recent literature has focused on the strongly positive biases in the polar region of the SO in DJF, we show here that the radiative biases for the Southern Ocean are complex and not always zonal. The summer time

position of the positive/negative biases agrees broadly with recent observational work analysing airmass characteristics (eg.

aerosol/airmass origin and impact on cloud condensation nuclei and cloud properties, Humphries et al., 2021; Simmons et al., 2021; Mace et al., 2021). This agreement highlights the importance of understanding not just the region where the biases are the most problematic, but the Southern Ocean system as a whole.

Rows 2-3 in Figure 1 show the respective biases for the CFL and CFIs compared to MODIS. Over all SO regions and seasons, negative CFL biases are found, in particular for the sub-polar and polar regions. The strongest bias is found in DJF for the polar region, though this extends into the sub-polar region, while the JJA appears to have the weakest biases. Interestingly, this bias in CFL is not compensated by a consistent overestimation in CFI for all regions (Figure 1, row 3). Annually, too much CFI is found for the polar and sub-polar regions, while this transitions to too little CFI in the mid-latitude region. In DJF, the positive sub-polar bias is very well defined and, as with the polar bias, at its largest. Over MAM, the sub-polar bias remains consistently constrained by the mid-latitude/sub-polar boundary, while the polar region bias begins to weaken and by JJA has become weakly negative. The clear distinction between the mid-latitude and sub-polar regions have also weakened in JJA, with the biases becoming less zonal. SON returns to predominantly positive polar and sub-polar regions, though with a less well defined boundary between the sub-polar and mid-latitude regions than what is seen for MAM or DJF.

Of note is the fact that the positive biases in the CFI are much weaker in magnitude than the negative biases in the CFL. Further, the spatial patterns of the biases are not the same. On average annually, the positive biases in the CFI over the polar and sub-polar regions only partially compensate for the negative biases in CFL. In the mid-latitude region however, both the cloud fraction biases (liquid and ice) are weakly negative. This indicates too few clouds overall in this region, which can explain the positive $SWCRE_{TOA}$ bias. Too few liquid clouds which are instead simulated as ice clouds, will result in clouds that are more optically thin causing too little short wave radiation to be reflected out to space. This may help explain the average positive $SWCRE_{TOA}$ bias in the polar region, however does not explain the more negative bias in the sub-polar region. The lack of spatial correlation for the cloud fractions indicates that although the inaccurate partitioning of liquid and ice cloud is an important contributor to the radiative bias, it is not the whole story.

Rows 4-5 in Figure 1 show the respective biases for the LWP and IWPs compared to MODIS. Broadly, the LWP and IWPs tend to be of opposite signs to each other and similar in magnitude, with positive LWP biases in the mid-latitudes, a transition to negative in the sub-polar region, and negative LWP in the polar region. However, the change from positive to negative (or vice versa for IWP) appears to be more southwards for the LWPs, while more northwards for the IWPs. Too much ice, in place of water, potentially super cooled liquid water, again would produce an optically thinner cloud, which would cause too much sunlight to reach the surface and a positive $SWCRE_{TOA}$ bias. In the polar region, this process is likely contributing to the positive $SWCRE_{TOA}$ bias. In the sub-polar region, in DJF and to a lesser degree the other seasons, both water paths are in places positive, increasing the optical thickness of the cloud overall, which may contribute to a negative $SWCRE_{TOA}$ bias. However, the regions of negative $SWCRE_{TOA}$ bias are not easily reconciled with the biases in the LWP and IWPs, indicating that this is a complex system. Finally, in the mid-latitudes region, a positive LWP bias is found over all seasons and a positive IWP in winter and spring and weakly negative IWP in summer and autumn. Positive water paths may have a compensating effect to the too little cloud fractions found in the region, resulting in the much weaker radiative bias. One reason for the

different biases in the warmer mid-latitude regions compared to the other regions is likely the temperature dependence with respect to super cooled liquid water formation.

To summarise, in the polar region, the frequency and LWP of liquid clouds is largely underestimated, likely resulting in a very strong positive radiative bias. This is compounded by a slight overestimation of ice clouds containing too much ice, which are optically thinner adding to the positive radiative bias. This finding agrees well with the literature in that not enough liquid water exists below zero degrees Celsius, instead being simulated as ice (Bodas-Salcedo et al., 2016b). In the sub-polar region, the frequency of liquid clouds is weakly underestimated, but is compensated by too much LWP, creating too few, optically thicker clouds. Combining these biases with too much ice cloud, and too much IWP, results in a weak radiative bias that fluctuates between positive and negative. Finally, in the mid-latitudes, negative CFLs are again combined with positive LWPs, while the CFI and IWP are both weakly negative resulting in a weak positive radiation bias. These patterns are found to be generally consistent across the seasons, with some degree of variability in the strength of the biases.

These results show that in the polar region, the biases in cloud fraction and water paths can satisfactorily explain the $SWCRE_{TOA}$ bias. For the other two regions however, the influence of the cloud biases is not as clear cut. The role of cloud phase, and in particular that of supercooled liquid water, appears to have significant latitudinal dependence, likely influenced by a range of factors, including temperatures and ice nucleating particle (INP) availability. This is a significant issue for model development, as has been shown previously, where fixing one of these issues for the SO region (e.g. the ratio of liquid to ice clouds, or number of INP) may have detrimental effects on other parts of the system. For this reason, we suggest that we need a more in-depth analysis of the problem with respect to the ACCESS-AM2 model, before we can attempt to reduce this $SWCRE_{TOA}$ bias. To do this, the next section presents results from the cloud regime-clustering, which will then be used to understand the radiative and cloud biases in more detail.

## 4 Cloud regimes

### 4.1 MODIS

The $k$-means clustering technique was applied to five years of MODIS daily-mean joint histograms over the entire globe. The 12 resulting cluster centres are shown in Figure 2. We have not shown the global relative frequency of occurrence (RFO), but have chosen to show the RFO for the Southern Hemisphere only in Figure 3, to focus on the Southern Ocean domain (30-69°S). The clusters have been approximately arranged from low to high CTP along the vertical and thin to thick $\tau$ in the horizontal.

Three of the resulting clusters are limited to the Antarctic region, and have similar (yet accordingly distinct) cloud characteristics. These clusters have come about in part as a result of our normalisation by one, instead of cloud fraction and make up only a small fraction of overall cloud occurrence. Furthermore, as noted by Williams and Webb (2009), cloud retrievals over ice-covered regions with high albedo can be problematic. For these reasons, the three Antarctic clusters are merged into one for the remainder of this analysis. We have labelled the remaining cloud regimes according to the cloud types that best reflect these profiles, however we acknowledge that the CTP-$\tau$ is often not enough to truly distinguish one cloud type from another

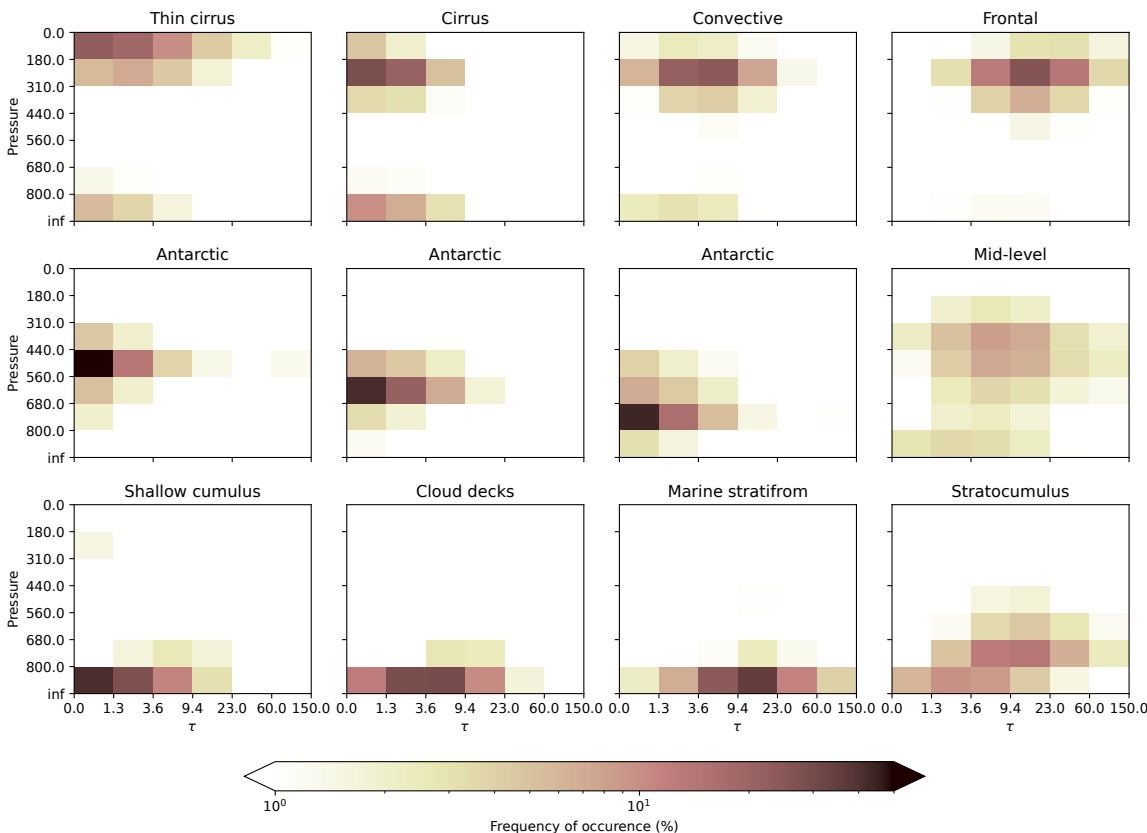

**Figure 2.** The mean CTP (y-axis, hPa) - $\tau$ (x-axis, optical depth unitless) histograms for each cloud regime found by $k$-means clustering on the MODIS product (daily means, 2015-2019). Each histogram has been assigned a 'cloud type'. The histograms have been loosely arranged by pressure and depth

(and that some quite different clouds may have similar CTP-$\tau$ profiles). Hence, these labels should be considered just that: a way of easily differentiating the cloud regimes in this study.

Along the top row of Figures 2 and 3, high level clouds are shown. The thin cirrus cluster is restricted to tropical regions, particularly over the West Pacific/East Indian Ocean, and is characterised by clouds that are optically thin with very low CTP. Similarly, the cirrus cluster is characterised by very optically thin clouds, at a slightly lower altitude, and is found predominantly in the tropical and mid-latitude regions. Both the cirrus and thin cirrus show some thin, low level cloud too, likely associated with shallow convective clouds. At a similar height, the convective cluster is more optically thick, and more strongly associated with the ITCZ and South Pacific convergence zone (SPCZ), although this is not shown by this map projection. Finally, along

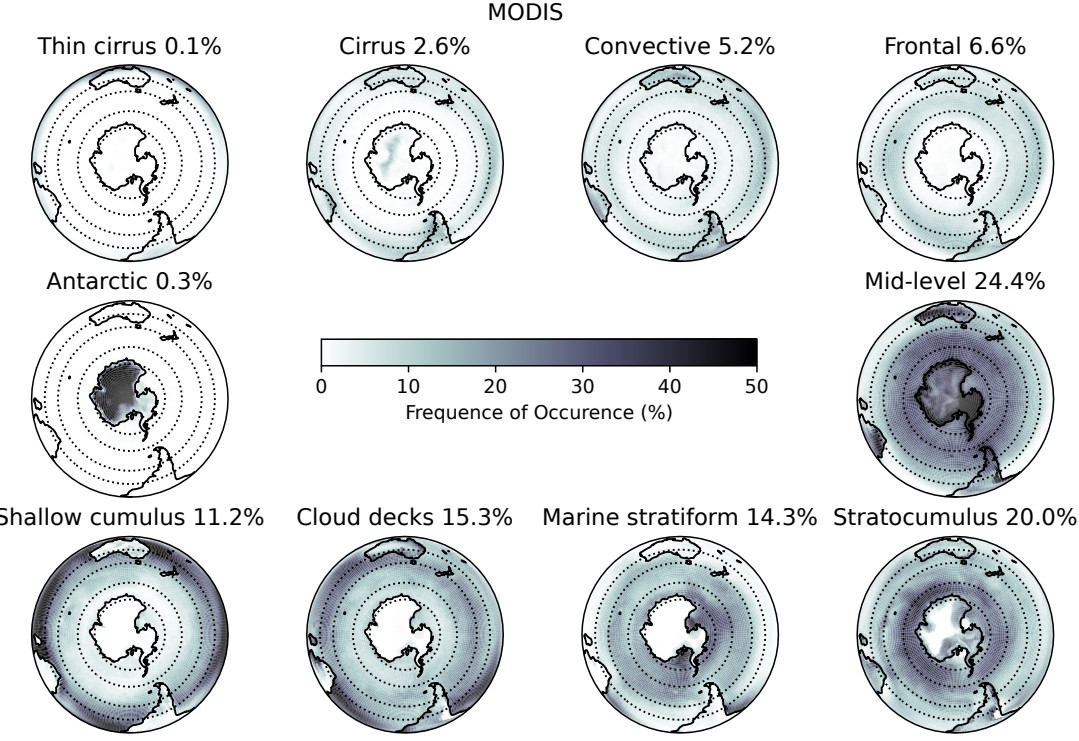

**Figure 3.** Spatial maps of the relative frequency of occurrence for each MODIS cloud cluster found by $k$-means clustering. Note that the three Antarctic clusters have been merged into one. The dashed lines represent the boundaries of the mid-latitude, sub polar and polar regions as defined in Section 3. The numbers in each title represent the mean frequency of occurrence for each cloud regime (in time and space) over SO region

330 the top row, the frontal cluster, while similar in altitude as the convective cluster, is more optically thick, and is found more predominantly in the mid-latitudes, in particular around westerly storm tracks. We note that convective and frontal clouds can have similar CTP-$\tau$ profiles (Williams and Webb, 2009), and hence their location is also important to consider.

The mid-level cluster shows mid-range optical properties as well as mid-level CTPs and is without one or two clearly dominant bins as found in the other clusters (Figure 2). This mid-level cluster is most common over the SO, marine equatorial
335 regions and terrestrial regions (in particular on the west coast of a number of continents). These varied geographic locations represent some significantly different cloud formation processes, raising an important point for this work: while the optical and height characteristics within a cluster may be similar, each cloud's trigger mechanism is not necessarily the same. While further analysis to understand the formation processes for the cloud regimes is possible via meteorological fields such as proximity to

cyclone centres or vertical motion, or physical fields such as proximity to topography would be of interest, such analysis is out of scope for this work.

Along the bottom row, the shallow cumulus cluster is found to have low, optically thin clouds that occur mostly over marine regions in the tropical and mid latitudes, away from any large-scale cloud formations such as the ITCZ/SPCZ (not shown) or stratiform cloud decks. This cluster has little vertical structure. By comparison, the cloud deck cluster is optically thicker and dominant in the eastern boundary of ocean basins, regions of extended stratiform cloud, again with little vertical structure. The marine stratiform cluster is the most optically thick cloud regime, with high CTPs and little vertical structure, and is found predominantly in higher latitude marine regions, as well as the very eastern boundaries of the cloud decks regions. Finally, the stratocumulus cluster characterised by mid-range optical thickness, but has lower CTPs than those of the other low cloud clusters, suggesting a greater vertical extent associated with stratocumulus clouds. This cluster is geographically wide spread, although dominant in the polar region of the SO and absent in regions of deep tropical convection.

## 4.2 ACCESS-AM2

Using the cluster centres defined by the MODIS joint histograms described above, we fit the ACCESS-AM2 joint histograms to the same 12 cluster definitions (noting that the three Antarctic clusters were subsequently merged). Figure 4 shows the difference in frequency of occurrence between ACCESS-AM2 and MODIS. It is immediately clear that ACCESS-AM2 has significant problems with the mid-level and stratocumulus clouds, simulating them too frequently (by 13.4% and 19.8%) across all regions. The low-level cloud fields by contrast are all underestimated, including shallow cumulus (-9.9%), cloud decks (-9.5%) and marine stratiform clouds (-10.0%). The higher clouds (thin cirrus, cirrus, convective and frontal), are simulated comparatively well, although also with slight underestimation of frequency. Interestingly, for the SO, most of the RFO biases are spatially consistent in sign. This result could be interpreted as a consistent bias, with consistent causes, across the latitudes. We will explore this in more detail in the subsequent sections.

## 5 Understanding the biases

### 5.1 Bias decomposition

We can now use the biases in the frequency of occurrence of the cloud regimes to help gain a better understanding of the biases in radiation using the traditional decomposition method described in Section 2.5. Note that from this point on we only consider the broad SO region, and the three sub-regions defined within. Figure 5 shows the cumulative components of the $SWCRE_{TOA}$ biases over each region, for each cloud regime. The horizontal bar represents the total bias. Similar to previous studies, including Bodas-Salcedo et al. (2014), Mason et al. (2015) and, Schuddeboom et al. (2018), the largest component of the $SWCRE_{TOA}$ bias is due to the errors in the RFO (blue bars). As previously discussed, DJF has the largest $SWCRE_{TOA}$ biases, though the dominance of the RFO bias is true for all seasons. For DJF, clear negative RFO errors are found for the mid-level clouds and the stratocumulus clouds, which are both overestimated in frequency. This negative RFO error has a

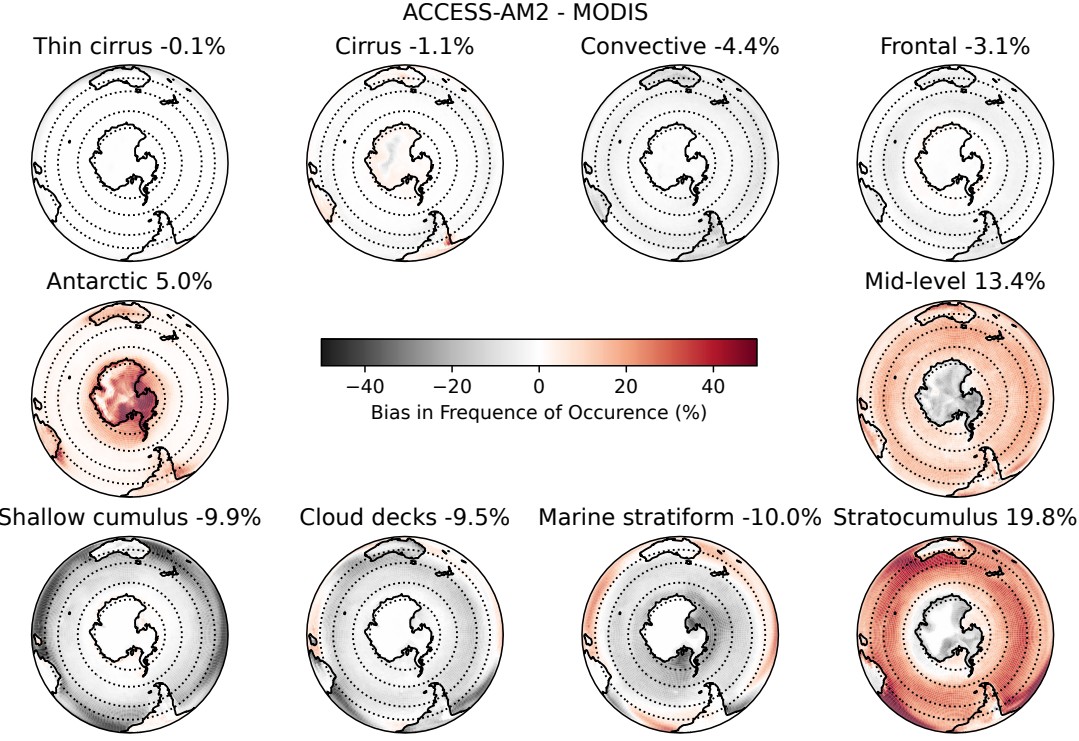

**Figure 4.** Spatial maps of the difference in the relative frequency of occurrence for ACCESS-AM2 - MODIS for each cloud cluster. The dashed lines represent the boundaries of the mid-latitude, sub polar and polar regions as defined in Section 3. The numbers in each title represent the mean frequency of occurrence for each cloud regime (in time and space) over SO region

compensating effect, as the mid-level clouds are simulated in the place of other, lower cloud regimes (eg. marine stratiform, cloud decks and shallow cumulus), which are underestimated in frequency and hence have positive RFO errors. A much smaller component of the bias is made up by the error in the simulated field (pink bars), primarily in DJF, while insignificant in the other seasons. Schuddeboom et al. (2018) hypothesised that the dominance of the RFO errors and comparatively smaller field errors, also found here, indicates that the clouds themselves, when simulated with the correct frequency, could in fact be simulated

well. This hypothesis will be tested in the next section. The cross term (green bars) appears to contribute to the radiative biases again only in summer, and predominantly in the polar region. What this cross term represents is not easy to evaluate, however it is clear that the field biases and RFO biases for this time and season can only explain part of the story.

We have also applied this bias decomposition technique to other cloud properties, including biases in LWP, IWP and CFI and CFL (see Appendix). The results for the LWP, IWP and CFL, for all regions, show large biases across all cloud clusters

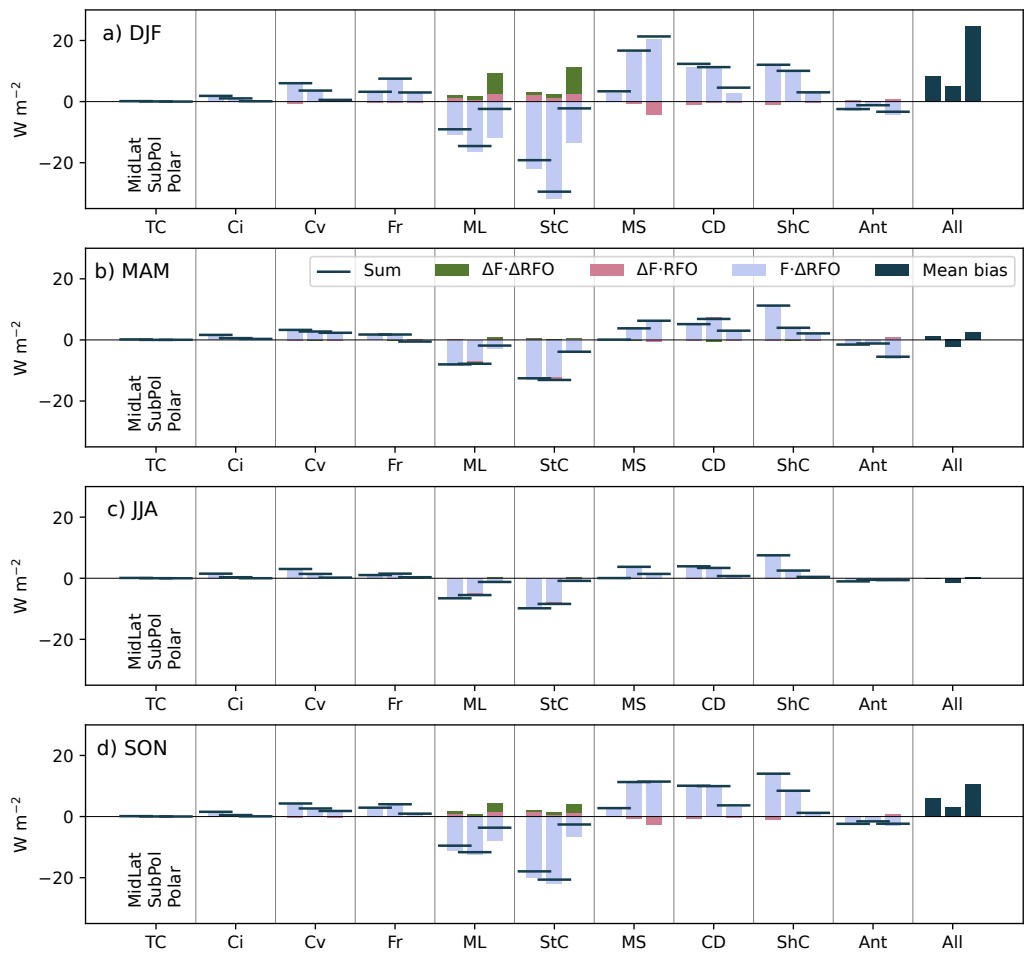

**Figure 5.** The decomposed mean biases in the $SWCRE_{TOA}$ ($W\,m^{-2}$) for each cloud regime (thin cirrus, TC; cirrus, Ci; convective, Cv; frontal, Fr; mid-level, ML; stratocumulus, StC; marine stratiform, MS; cloud decks, CD; shallow cumulus ShC; Antarctic, Ant) over the three regions from left to right mid-latitudes, sub-polar and polar, for each season (DJF, plot a; MAM, b; JJA, c; SON, d respectively). The sum of the decomposed biases are shown by the horizontal bar, while the the terms of the bias decomposition (see eq. 1) are shown in blue ($F_r^{sat} \cdot \Delta RFO_r$), pink ($\Delta F_r \cdot RFO_r^{sat}$) and green ($\Delta F_r \cdot \Delta RFO_r$). On the far right of each plot is the mean radiative bias for each region/season as a reference

and decomposed terms. Interestingly, the field error and RFO error are of opposite signs and approximately cancel, leaving the cross term to make up the total radiative bias. For CFI, the RFO error was dominant, similar to the $SWCRE_{TOA}$, though with the field error and cross term being non-negligible, unlike for the $SWCRE_{TOA}$. The importance of the cross term in these results makes the respective biases much more difficult to understand, though it is clear that the decomposed errors in RFO and the fields themselves are playing a large role in the total bias.

The strength in this method is that it is time and space agnostic, i.e. it does not require the two 'climates' to be directly comparable, as it is using the mean RFOs and field values to make judgements about how each of them contribute to a particular bias. For this reason, this method has been popular in the literature which has often used CMIP simulations as the basis of their work. However, a limitation of this method is that it cannot compare like-for-like instances in cloud properties to gain a better understanding of what particular conditions are leading to large $SWCRE_{TOA}$ biases. This work suggests

that for all seasons and all regions, a negative bias occurs because the frequency of mid-level and stratocumulus clouds are overestimated, compensating for the underestimation of the lower level clouds. We ask, at what cost, specifically, were the mid-level and stratocumulous clouds overestimated? Is the compensating error due to the model predicting clouds that are too frequent, optically thick, high or with too much liquid? Applying the same technique (the bias decomposition) to such cloud fields was unable to satisfactorily give us these answers. Additionally, we would like to know, does the model ever actually do

a good job, with the correct cloud type and a small (or wishfully - no) radiative bias?

### 5.2   Biases when clusters are correctly or incorrectly simulated

One strength of comparing a daily, nudged, simulation to daily MODIS fields is the ability to make direct comparisons in time and space. The synoptic meteorology is considered to be the same in the model and the observed conditions, due to the nudging of the model. We therefore expect that the model microphysics, if accurate, would generate the same cloud types that

the large-scale dynamics prescribes. With this assumption, we are able to isolate instances (exact points in time and space) where the model simulates the same cloud type as MODIS, which we define as 'correctly' simulating the cloud type. Similarly, we can also define the instances where the model simulates a different cloud type, which we consider an 'incorrect' cloud type assignment by the model. We can then use these instances to generate a statistical understanding of the conditions during these times and places. We demonstrate these definitions in Figure 6a and b.

In the previous section, we saw that the model generally tends to simulate the cloud type RFOs incorrectly and that this bias in RFOs dominates the radiative bias. This result demonstrates the limit of the decomposition method as we are not able to gain any further information about how the microphysical biases in cloud types affect the radiative properties, or the nuances of which cloud types are being simulated in place of others. In this Section, we take advantage of the model nudging and explore in detail the instances when ACCESS-AM2 correctly and incorrectly simulates the MODIS identified cloud regimes and begin

to understand the associated radiative and cloud biases in much greater detail than has previously been achieved.

Each panel of Figure 7 shows for each MODIS cloud type (y-axis), the percent of time that each cloud type is assigned by ACCESS-AM2 (x-axis). The total number of instances that that cloud type is observed by MODIS for each region is shown on the right of each panel. If ACCESS-AM2 simulated every cloud type the same as what the MODIS product did, we would

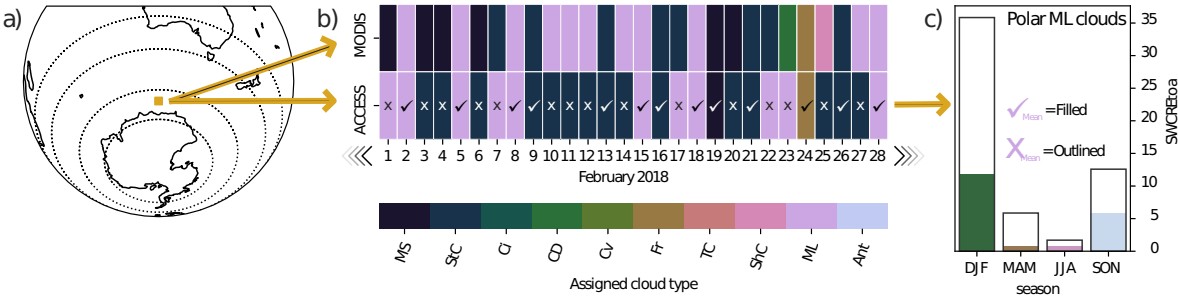

**Figure 6.** For each grid box (a), and for every point in time (b), the cloud type has been determined via k-means clustering for the MODIS product, and subsequently fitted to the ACCESS-AM2 output. The assignment of cloud type is shown for a one month period in b), where we can see where ACCESS-AM2 correctly (shown by a tick) or incorrectly (shown by a cross) simulates the same cloud type as the MODIS product. Using this information, we calculate the seasonal mean radiation or cloud property bias (c) over the instances where ACCESS correctly (filled bars) or incorrectly (outlined bars) simulated the cloud type for the three regions of the Southern Ocean. As an example, c) shows the mean radiation biases for polar mid-level clouds.

expect a diagonal line through each panel in Figure 7 of 100%, with zeros elsewhere. What is shown however, reflects clearly
the biases of the RFOs, where ACCESS incorrectly assigns the stratocumulus and mid-level clusters over most other clusters. For the low-level cloud regimes (shallow cumulus, cloud decks and marine stratiform), we see that a large number (between 48% and 59%) of points have been wrongly assigned to the stratocumulus cluster, while the high level clouds (convective and frontal, as well as the less important cirrus and thin cirrus) tend to be assigned as the mid-level cluster (between 38% and 67%). Only 12%, 11% and 5% of the marine stratiform points have been correctly assigned by ACCESS-AM2 for each region
(mid-latitudes, sub-polar and polar). This statistic is worse for some other, less important SO cloud types. In contrast, 63%, 68% and 68% of mid-level clouds and 60%, 66% and 56% of stratocumulus clouds are simulated in the correct time and place for the three SO regions (mid-latitudes, sub-polar and polar).

Having identified exactly when ACCESS-AM2 correctly or incorrectly simulates cloud regimes, we can now begin to evaluate what the biases in radiation and other cloud fields are in these instances. Figures 8 (mid latitude), 9 (sub-polar) and 10
(polar), show seasonal mean biases when the ACCESS-AM2 correctly simulates each cloud cluster (coloured bars) or incorrectly simulates each cloud cluster (black transparent bars). An example of how one of these subsets (polar mid-level clouds) is derived is shown in Figure 6. We note that these results are not weighted by the frequency of occurrence, unlike the previous figure, as we are interested in understanding the microphysical properties of each cloud type, even if they occur infrequently. Figure 7 can provide an indication of the frequency of each cloud type and how often it is correctly or incorrectly identified.
Furthermore, we emphasise that these plots show the impact in the radiation and cloud fields from the perspective of the 'right' cluster being assigned wrongly, unlike in Figure 5, which shows the radiative biases from the perspective of the accumulative 'wrong' clusters. For this reason, some of the values between the figures have opposite signs, but support the same overall finding.

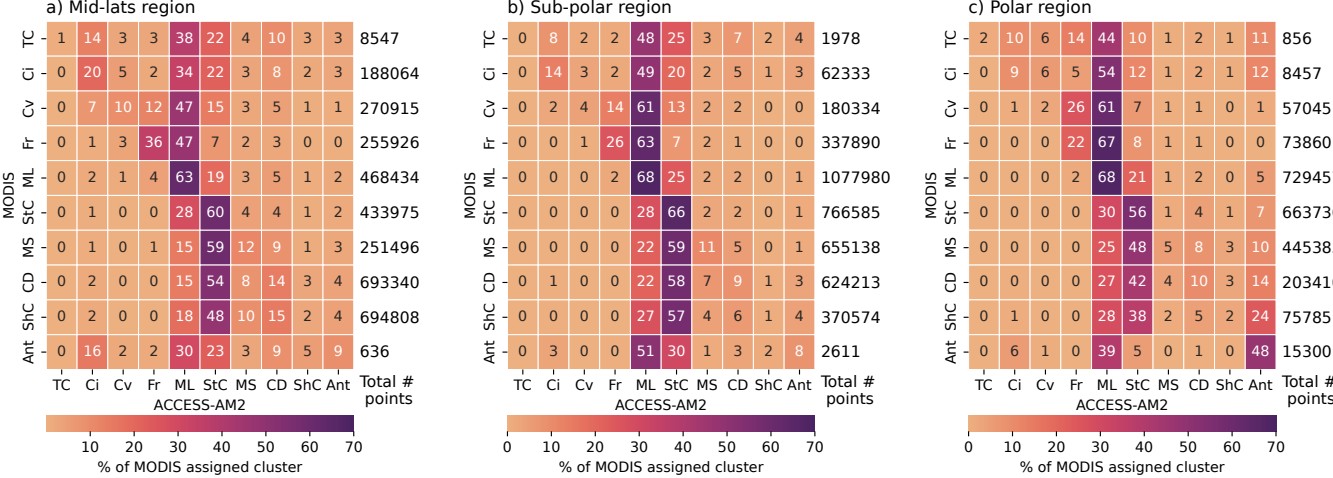

**Figure 7.** Confusion matrices for each region, a) mid-latitudes, b) sub-polar and c) polar, showing the MODIS cluster assignment on the y-axis and the ACCESS-AM2 cluster assignment on the y-axis (thin cirrus, TC; cirrus, Ci; convective, Cv; frontal, Fr; mid-level, ML; stratocumulus, StC; marine stratiform, MS; cloud decks, CD; shallow cumulus ShC; Antarctic, Ant). The colours (and inset text) represent the proportion of points assigned to each MODIS cluster (eg. sum to 100 along the x-axis). The text on the right hand side indicates the number of points (in time and space) that make up these statistics for each MODIS cluster

### 5.2.1 The mid-latitude region

Figure 8a shows the mid-latitude region's mean $SWCRE_{TOA}$ biases when ACCESS-AM2 correctly and incorrectly simulates the cloud types. For the mid-level and marine stratiform clouds, the $SWCRE_{TOA}$ biases are larger when the cloud types are assigned incorrectly (outlined bars) than when they have been correctly simulated by ACCESS-AM2 (coloured bars). However, when these cloud types are correctly assigned (which happens 63% and 12% of the time respectively), the $SWCRE_{TOA}$ bias are also non-negligible in most seasons. For the mid-level clouds, the $SWCRE_{TOA}$ biases are smaller in magnitude when the clusters are correctly identified by the model. Interestingly, this is not the case for the CFL and LWPs, which both have larger biases when the clusters agree, suggesting that the radiative effects associated with too few liquid water clouds is partially compensated by them being too optically thick. This indicates that the lower $SWCRE_{TOA}$ bias when the mid-level clouds are correctly simulated may be occurring for the wrong reasons. For the marine stratiform clouds, the CFL is strongly underestimated when it is incorrectly simulated, which is expected to produce positive $SWCRE_{TOA}$ biases. The CFI is comparatively well simulated. Interestingly, the LWP and IWPs seem to be relatively well captured for marine stratiform clouds. This suggests that the $SWCRE_{TOA}$ bias may be predominantly driven by an underestimated CFL, while the amount of water in them is somewhat correct.

The stratocumulus cluster shows almost uniformly equal biases in all fields when the cluster assignments are correct and incorrect. The positive radiative biases are predominantly associated with too few liquid water clouds, similar to that of the mid-level clouds. Considering that stratocumulus clouds are simulated correctly 60% of the time and are otherwise simulated

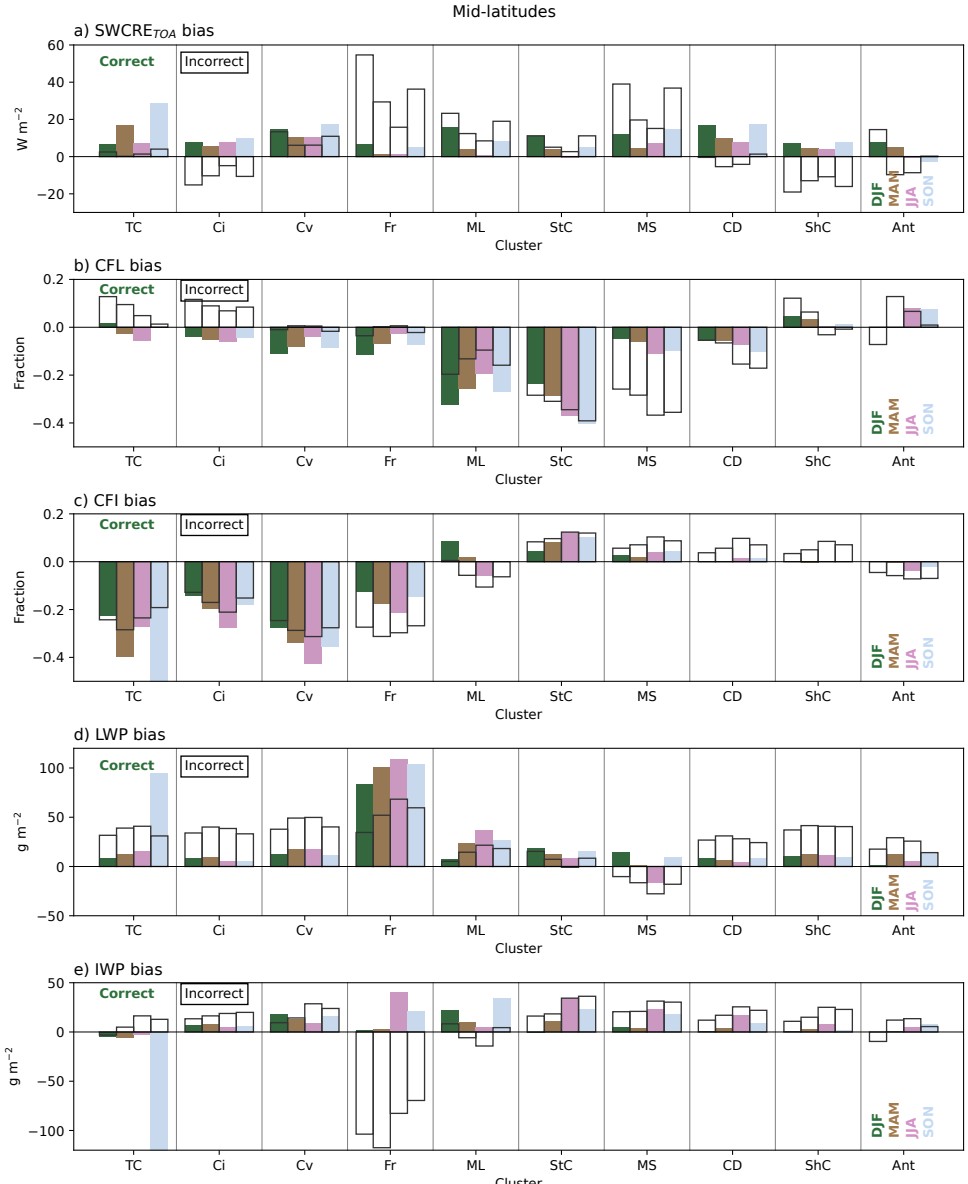

**Figure 8.** a) Mid-latitude SWCRE$_{TOA}$ biases ($W\,m^{-2}$) averaged over each cloud cluster (thin cirrus, TC; cirrus, Ci; convective, Cv; frontal, Fr; mid-level, ML; stratocumulus, StC; marine stratiform, MS; cloud decks, CD; shallow cumulus ShC; Antarctic, Ant), seasonally (DJF, dark blue; MAM, green; JJA, pink; SON, light blue), for instances when the MODIS clusters are correctly assigned by ACCESS-AM2 (coloured bars) and are incorrectly assigned (black outline) for the mid-latitude region. b-e) shows the same as plot a) but for the CFL (fraction), CFI (fraction), LWP ($g\,m^{-2}$) and IWP ($g\,m^{-2}$) biases respectively.

as mid-level clouds 28% of the time, these biases make sense. These results suggest that we cannot focus solely on improving the lowest level clouds, but must also improve the mid-level cloud representation too.

Cloud decks present a notable exception of where the radiative biases are larger and of opposite sign when the cluster is correctly identified. Cloud decks are not simulated frequently enough in ACCESS-AM2 (correct only 14% of the time), instead assigned as stratocumulus clouds 54% of the time. Figure 8 shows a weak negative $SWCRE_{TOA}$ bias for the MAM and JJA seasons when the ACCESS-AM2 clusters are incorrect, while the bias in the other seasons is negligible. When cloud-decks are correctly assigned, the $SWCRE_{TOA}$ bias is positive in all seasons, indicating that for these low-lying clouds, even if we are able to simulate the height and optical thickness properties of the cloud correctly, issues remain. Looking at the cloud fields, most show small biases (compared to the other cloud regimes), with the CFL having the largest negative bias. We suggest that too low cloud fraction would result in a positive $SWCRE_{TOA}$ bias, though how this seemingly minor cloud fraction bias is resulting in the more significant radiative bias is difficult to say. Regardless, this result demonstrates the complexity of this issue, where we not only need to get the right clouds for the large-scale conditions, but also ensure the underlying microphysical properties are correctly simulated.

Shallow cumulus clouds show a small positive $SWCRE_{TOA}$ bias when the ACCESS-AM2 correctly simulates the cluster, which is also reflected in the cloud fraction and water paths, indicating that the model performs well in these instances. However, Figure 7a shows that the model infrequently gets this cloud type right (only 2% of the time), instead simulating it as stratocumulus 48% of the time in the mid-latitudes. The negative $SWCRE_{TOA}$ bias found in all seasons when shallow cumulus are incorrectly simulated is likely a reflection of the higher and more optically thick clouds found in the stratocumulus cloud type, indicated by overestimated LWP and to a lesser degree, overestimated IWP. This result also agrees well with our previous plot (Figure 5), where too many stratocumulus clouds result in a compensating negative bias. A similar result can be observed for the less frequent cirrus clouds, which are too often simulated as lower, more optically thick stratocumulus or mid-level clouds (22% and 34% of the time) and cloud decks in winter and autumn.

The frontal cloud type has a large positive $SWCRE_{TOA}$ bias when the model does not correctly simulate the MODIS cluster, which occurs 64% of the time in the mid-latitude region. The frontal clouds are a relatively high cloud (see Figure 2) and in these lower latitudes have a greater likelihood of being ice clouds. This bias, found across all seasons, could be due to a too large LWP, though the CFL is relatively well captured. For the frontal ice clouds, too few clouds are simulated with a negative IWP bias. In the case of correctly assigned frontal clouds, we can see that the $SWCRE_{TOA}$ bias is small, despite too little CFI and too high LWP. This result may again indicate that while the ACCESS-AM2 model is simulating correct $SWCRE_{TOA}$, it may be doing so for the wrong reasons. A similar result, i.e. strongly underestimated CFI causing a positive radiative bias, is found for the convective clouds, in instances when they both disagree and to a greater extent, agree. This result may indicate that for these higher clouds, getting the cloud fractions right, with the right phase partitioning, may go a long way to reducing the radiative biases. The thin cirrus clouds are shown to have large biases in the spring (SON) when correctly simulated for all fields. However, we note that these cloud types occur infrequently (see Figure 7a) and are only correctly simulated 1% of the time, and hence, we do draw any conclusions from this result.

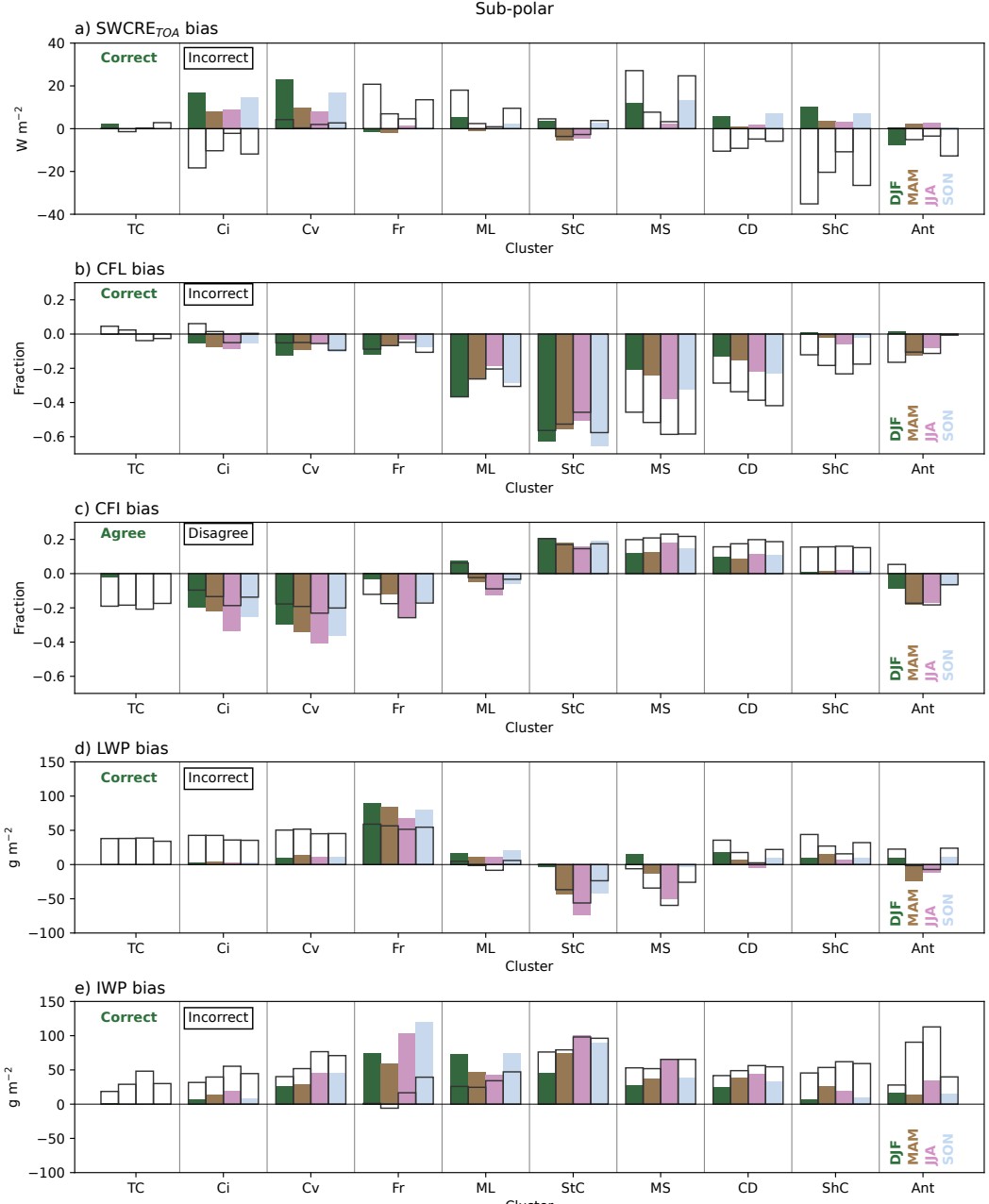

**Figure 9.** The same as Figure 8, but for the sub-polar region.

### 5.2.2   The sub-polar region

The sub-polar total radiative bias, shown in the far right column of Figure 5, is in general quite small. If we consider just the zonal means, the model appears to perform well. However, after analysing Figure 1, we know that large spatial variability exists within this region that leads to a small overall bias. In this section we explore if some of that variability can be attributed to particular cloud types.

The mid-level and stratocumulus clouds are the two most accurately simulated cloud types (68% and 66% of the time). The cloud radiative biases in these cases are relatively low, implying that the model may be doing a good job. We note that the biases are fairly similar for when they disagree too, and that is likely due to each one being predominately assigned as the other when they are incorrectly simulated. When the stratocumulus are correctly simulated, examination of the cloud fields indicates that despite the small radiative biases the apparent model skill may be misleading. In these cases, CFL and LWP are underestimated and the CFI and IWP are overestimated across most seasons. We suggest that too few, optically thinner (derived from too little LWP) liquid clouds, causing a strong positive $SWCRE_{TOA}$ bias may be partially compensated by too many, optically thicker (too much IWP) ice clouds. A similar result is found for the mid-level clouds, though with smaller biases in CFL and IWP, and negligible biases in CFI and LWP.

We can see that the majority of the negative bias in the sub-polar region is driven by shallow cumulus clouds that are incorrectly simulated. Shallow cumulus clouds are optically thin, and low lying (see Figure 2). They are only correctly simulated in this region 1% of the time. Instead they are simulated as stratocumulus clouds 57% of the time which are higher and more optically thick, both of which would lead to more sunlight being reflected back out to space. Again, this agrees well with the findings of Figure 5, where negative radiative biases are found to be due to too many stratocumulus and mid-level clouds. Looking at the shallow cumulus cloud properties, we can see that when these clouds are incorrectly defined, we have too few liquid clouds, too many ice clouds and too high LWP and IWPs. When shallow cumulus are correctly simulated, the majority of these biases are much smaller, and the $SWCRE_{TOA}$ bias is weakly positive, indicating again that the model is doing a better job.

The cloud deck clusters, when incorrectly assigned, also appear to contribute towards a negative $SWCRE_{TOA}$ bias. Cloud decks are only correctly simulated 9% of the time, and are also considered by the ACCESS-AM2 model to be stratocumulus clouds 57% of the time instead. Examining the biases in the cloud fields for these instances indicates a similar process as for the shallow cumulus clouds for the $SWCRE_{TOA}$ bias to occur. For the sake brevity, we will not discuss the remaining cloud types for this region. The majority of them (bar some of the less frequently occurring types) are generally contributing to a positive $SWCRE_{TOA}$ bias regardless of whether the clusters are correctly identified or not.

### 5.2.3   The polar region

Finally, we now consider the polar region in Figure 10. Here it is clear that the mid-level, stratocumulus and marine stratiform clouds are contributing the most to the positive $SWCRE_{TOA}$ bias, whether the clusters are correctly or incorrectly assigned. Few compensating errors are found in cloud types with a meaningful impact on the region (e.g. that occur frequently), adding

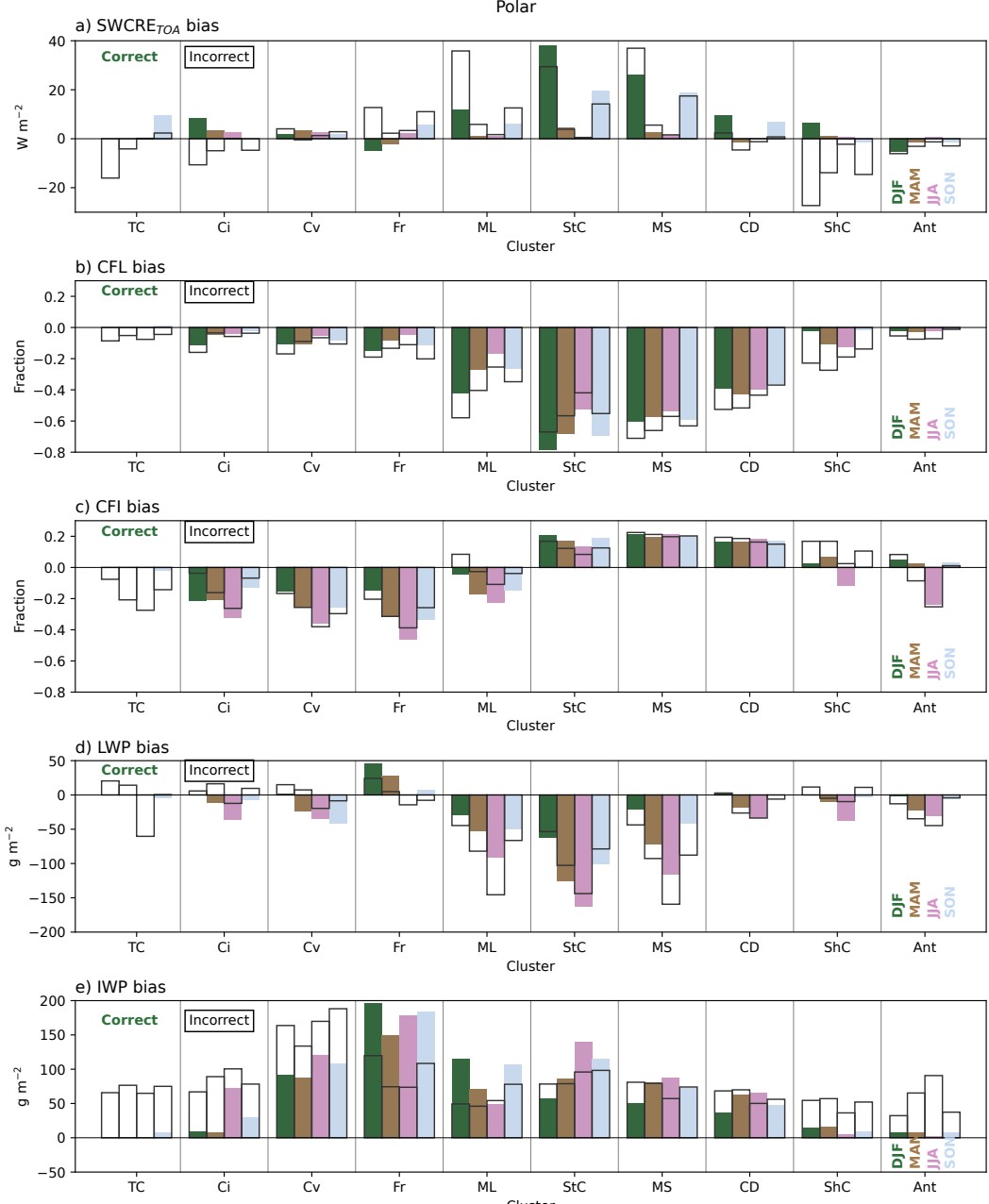

**Figure 10.** The same as Figure 8, but for the polar region.

to the larger magnitude of the SWCRE$_{TOA}$ bias in this region. In each of the three aforementioned cloud types, CFL is strongly underestimated (by over 50% in most seasons) by the model. The LWP is predominantly underestimated (though interestingly, less so for summer) and the IWP is overestimated. Each of these cloud biases are expected to produce a positive SWCRE$_{TOA}$

bias. The CFI, for the marine stratiform and stratocumulus clouds is overestimated, while for the mid-level clouds, it is in most cases underestimated. This difference may be due to (in part) the higher CTPs found in these mid-level clouds, meaning that ice formed in the model is done so correctly. The cloud biases found in these plots are of similar magnitude throughout the year while the radiative bias in summer time is large and almost non-existent in winter. This is due the fact that despite consistently poorly simulated cloud properties, they have much lesser impact on the radiative balance in winter due to the much smaller amount of solar radiation compared to the summer. Nevertheless, the fact that even when the cloud types are correctly simulated by the ACCESS-AM2 model the SWCRE$_{TOA}$ bias are similarly, if not more, positive than when the clusters are incorrectly assigned is a cause for concern. These results strongly highlight the issues of cloud phase within the model, that exists even if the height and optical depth of the cloud is correctly simulated. Furthermore, these results show that the cloud property biases are not limited to the summer season, where they are able to have a large impact on the SWCRE$_{TOA}$.

### 5.2.4 Summary of findings

There are two broad groups of cloud types driving the SWCRE$_{TOA}$ bias apparent from this analysis: the high clouds, including thin cirrus, cirrus, convective and frontal clouds; and mid-low level clouds, including the mid-level, stratocumulus, marine stratiform, cloud deck and shallow cumulus clouds. The high clouds consistently, across regions and seasons, contain too few ice clouds (negative CFI) with too much IWP. The IWP biases get larger at higher latitudes. While these cloud types do not make up a large fraction of clouds over the SO (see Figure 3), and hence have not been widely studied, it is clear that the different microphysical processes are also contributing to a SO positive SWCRE$_{TOA}$ bias.

For the mid-low level clouds, too few liquid clouds are found consistently across the regions and seasons, which again, gets worse at higher latitudes. This contribution to the positive SWCRE$_{TOA}$ bias is compounded in the optically thicker cloud types (mid-level, stratocumulus and marine stratiform) by negative biases in the LWP, also increasing in magnitude with higher latitudes. Positive biases in CFI and IWP are also found for these cloud types, which again are largest in magnitude in the polar region. These positive biases in ice clouds may have a compensating negative effect to the SWCRE$_{TOA}$ bias if they are increasing the overall cloud and water path fractions, however, if they are simulated instead of super cooled liquid clouds, they may indeed have the opposite effect. The optically thinner shallow cumulus clouds are found to have a compensating negative effect when the optically thicker clouds are simulated in their place for all regions.

### 6 Conclusions

Recent evaluation of CMIP6 models has found continuing problems in the SO shortwave cloud radiation effect. While updates to model parameterisations and tuning have occurred since CMIP5, these have resulted in some models having too high climate sensitivity (Zelinka et al., 2020), particularly driven by the SO, that may be producing lower SO SWCRE$_{TOA}$ biases for the wrong reasons (Schuddeboom and McDonald, 2021; Gettelman et al., 2020). In this work we take a detailed look at the SWCRE$_{TOA}$ bias in one model, the ACCESS-AM2 model. By running ACCESS-AM2 with nudging, we are able to make

day-for-day comparisons with satellite products (in this case CERES-Syn1D and MODIS COSP products) for the first time, allowing more in-depth analysis of this problem than what has previously been completed.

In this analysis, we use unsupervised $k$-means clustering on CTP-$\tau$ histograms, a method that has proven useful in numerous previous studies. We use 12 clusters for our analysis, though later merge three in to one 'Antarctic' cluster, leaving ten cloud regimes from which we can begin to understand under what conditions the SWCRE$_{TOA}$ bias occurs and the associated cloud properties. In particular, we are able to analyse the SWCRE$_{TOA}$ and cloud property biases in instances when the model correctly simulates (e.g. hits) a cloud regime and compare it against instances when the model incorrectly simulates a cloud regime (e.g. misses - assigned as something else) with respect to the MODIS product.

We find that ACCESS-AM2 strongly overestimates the occurrence of stratocumulus and mid-level clouds over the entire SO, in general agreement with that found for CMIP6 models in Schuddeboom and McDonald (2021). The over-prediction of these two cloud types comes predominantly at the cost of marine stratiform, cloud deck or shallow cumulus clouds (and others), while stratocumulus and mid-level clouds themselves are accurately predicted 60% and 67% of the time by the model, respectively. In particular, we find shallow cumulus clouds are simulated as the higher/thicker stratocumulus clouds by ACCESS-AM2 50% of the time, leading to a compensating negative SWCRE$_{TOA}$ bias over the entire SO for all seasons. A similar result is found in the sub-polar region for cloud decks in all seasons. Interestingly, when shallow cumulus clouds are correctly simulated by ACCESS-AM2, the radiative biases are much smaller, as are the biases in cloud properties. This result implies that correctly simulating the conditions for shallow cumulus to form may have a beneficial impact on the simulated radiative balance. The same cannot be said for the cloud deck clouds however, where errors in both the SWCRE$_{TOA}$ and cloud properties remain even if they are correctly simulated.

We find that when stratocumulus clouds are correctly simulated by the ACCESS-AM2 model, the SWCRE$_{TOA}$ bias in the mid-latitude and sub-polar parts of the SO is relatively small. However, examination of the respective biases in cloud properties indicates that the model is likely producing these smaller radiative biases for the wrong reasons, which differs by latitude. Concerningly, in the polar region summer time, stratocumulus clouds have a worse SWCRE$_{TOA}$ bias when they are correctly simulated by the ACCESS-AM2 model than when they are incorrectly assigned, indicating that significant issues remain within cloud microphysical properties, and in particular cloud phase, even if the macrophysics are somewhat correct. These findings provide important knowledge to help guide future model development which must target not only simulating the correct cloud type but also the correct cloud microphysics.

Marine-stratiform, stratocumulus and mid-level clouds contribute most to the polar region's positive SWCRE$_{TOA}$ bias. Each of these cloud types is associated with large biases in cloud properties, including too few liquid clouds, too little LWP and too much IWP. This is the case both for when these cloud types are correctly and incorrectly assigned. In the case of the marine-stratiform and stratocumulus clouds, the CFI is overestimated, while for the mid-level clouds, it is underestimated in all seasons except summer, where the bias is small. These results indicate that the issue of cloud phase within the ACCESS-AM2 model is still causing significant problems for the polar radiative balance. This cloud phase problem in these three cloud regimes does not appear to be as large an issue for the same clouds in the mid-latitudes or even the sub-polar region. This finding implies

that any changes to the model parameterisations must be done with caution, taking into account latitudinal dependencies on things such as temperature, boundary layer coupling, ice nucleating particle presence, etc.

While we have in this work explored the $SWCRE_{TOA}$ biases between ACCESS-AM2 and the CERES-Syn1D product and the cloud feature and type biases between ACCESS-AM2 and MODIS product, some key caveats must be raised. Firstly, we must note that comparing across two different satellite products may not be ideal given different assumptions and requirements. While this is standard practice in the field, the two products are derived from different platforms with different limitations, pointing towards a greater need for combined 'earth system' satellite products. We note that other satellite products have not been used in this work due to inconsistencies with the time period of focus. Secondly, we must recognise that both satellite products have their own biases. While in this work we regard them as 'truth', this is not necessarily the case. Hinkelman and Marchand (2020) for example, compared CERES and CloudSat radiation fields to in-situ observations at Macquarie Island and found a $+10\,W\,m^{-2}$ bias, which they expect would exacerbate the biases found in model evaluation. As discussed in Section 2.2, significant issues remain with the retrievals of $R_{eff}$, and subsequently the water paths. The propagation of these errors through the COSP framework, meant that comparisons between the COSP product and MODIS product were found to be very unrealistic, and hence these fields were not used and the raw model field LWP and IWP were used instead, adding uncertainty to these results. Similarly, having to convert between in-cloud and grid-box mean water paths has added another layer of uncertainty to this work. We chose to use the grid-box mean values, but the use of in-cloud values would have changed the our results and their interpretation. Whilst the COSP framework and specifically derived MODIS product go a long way to helping model evaluation, more needs to be done to ensure that modellers can use an observational products with confidence and minimal transformation. Further work evaluating cloud properties derived from satellite products would be of upmost help to studies such as this.

Gettelman et al. (2020) also noted that the nudging choices made can have large, and in some cases detrimental effects on the ability to simulate cloud fields, in particular cloud phase partitioning and water content due to the control on temperatures Zhang et al. (2014). While a study such as this could not be performed without the use of nudging, it is possible that the effects found in Gettelman et al. (2020) may be present here. We must also consider the impact of a single moment cloud scheme used in ACCESS-AM2 (the PC2 scheme). A double moment cloud scheme, that includes prognostic equations for both the cloud droplet size and number is preferable for a study such as this. Even better would be a cloud scheme that is fully coupled to the aerosol scheme. While in ACCESS-AM2, the cloud droplet number concentration is resolved, ice nucleating particles are not explicitly resolved. We hope to be able to address both of these issues in the future within ACCESS.

The work presented in this study has provided a relatively qualitative view of the cloud and radiative biases associated with cloud types. Studies such as Bender et al. (2017) have used more quantitative methods to evaluate the role cloud fractions play in determining the radiative balance. Specifically, Bender et al. (2017) examine the distributions of cloud albedo and the associated cloud fraction in CMIP5 models and their linear relationships. In further exploration of the data used in this work, we note that most of the relationships between cloud properties and radiative bias are in fact non-linear, which is also highlighted in this work by the differing relationships observed by latitude/cloud type. Similar results have been noted in Bodas-Salcedo et al. (2016a), where the LWP and radiative biases were not found to be as tightly coupled as expected. We will present findings

of a quantitative analysis employing machine learning to understand the role cloud properties play in determining the radiative bias in an upcoming paper.

Furthermore, in this work we have only considered a few cloud properties that impact the radiation budget. Other important factors, such as the number and size of cloud droplets, precipitation phase and amount, or other thermodynamical properties are likely to impact the absorption and scattering properties of clouds and their lifetime (for example: Mülmenstädt et al., 2021). Additionally, we have only considered these properties in isolation, eg. one field at a time, and not how they are operating together. This will also be addressed in our upcoming machine learning publication.

To summarise, in this work we have found that considerable radiative biases continue to exist within the ACCESS-AM2 model. By analysing this bias with respect to cloud types and their properties, we find that significant issues with respect to cloud phase remain in the mid-low level cloud regimes for the polar region of the SO, regardless of season, and that even if these cloud types are correctly simulated by the model, the large microphysical biases still persist. We also find compensating errors due to the underestimation of shallow cumulus clouds in favour of stratocumulus clouds. Our results show that significant effort must continue to reduce these SO cloud biases within models.

*Code and data availability.* Limited model data for this project is available at https://doi.org/10.5281/zenodo.6004062. The code to perform the analysis of this work can be found on Github (accessed 28 June 2022). The model set-up (under licence) can be found by UM users under suite bx400. CERES data can be downloaded from https://ceres.larc.nasa.gov/data/ (last accessed 25th March 2022). MODIS data can be downloaded from https://ladsweb.modaps.eosdis.nasa.gov/archive/allData/61/MCD06COSP_D3_MODIS/ (last accessed 25th March 2022).

*Author contributions.* SF completed the model simulations, analysis and writing of this work. AP, MM, SA and MW contributed to the planning, ideas and revisions of this paper. MW provided guidance on the model setup

*Competing interests.* The authors have no competing interests to declare

*Acknowledgements.* This project received grant funding from the Australian Government as part of the Antarctic Science Collaboration Initiative program, under the Australian Antarctic Program Partnership, ASCI000002. This research was undertaken with the assistance of resources and services from the National Computational Infrastructure (Project jk72), which is supported by the Australian Government. S.F. would like to thank Scott Wales, and the ARC Centre of Excellence for Climate Extremes for their maintenance of virtual environments and code/model support. S.F. would also like to thank Robert Pincus and Alejandro Bodas-Salcedo for early advice on the use of MODIS/COSP products. The authors would like to acknowledge the teams at NASA CERES, NASA Earth data and the CFMIP project for making the tools and data used in this work available. The contribution of S.P.A. was supported by the Australian Antarctic Science project 4292

## Appendix A: Additional figures

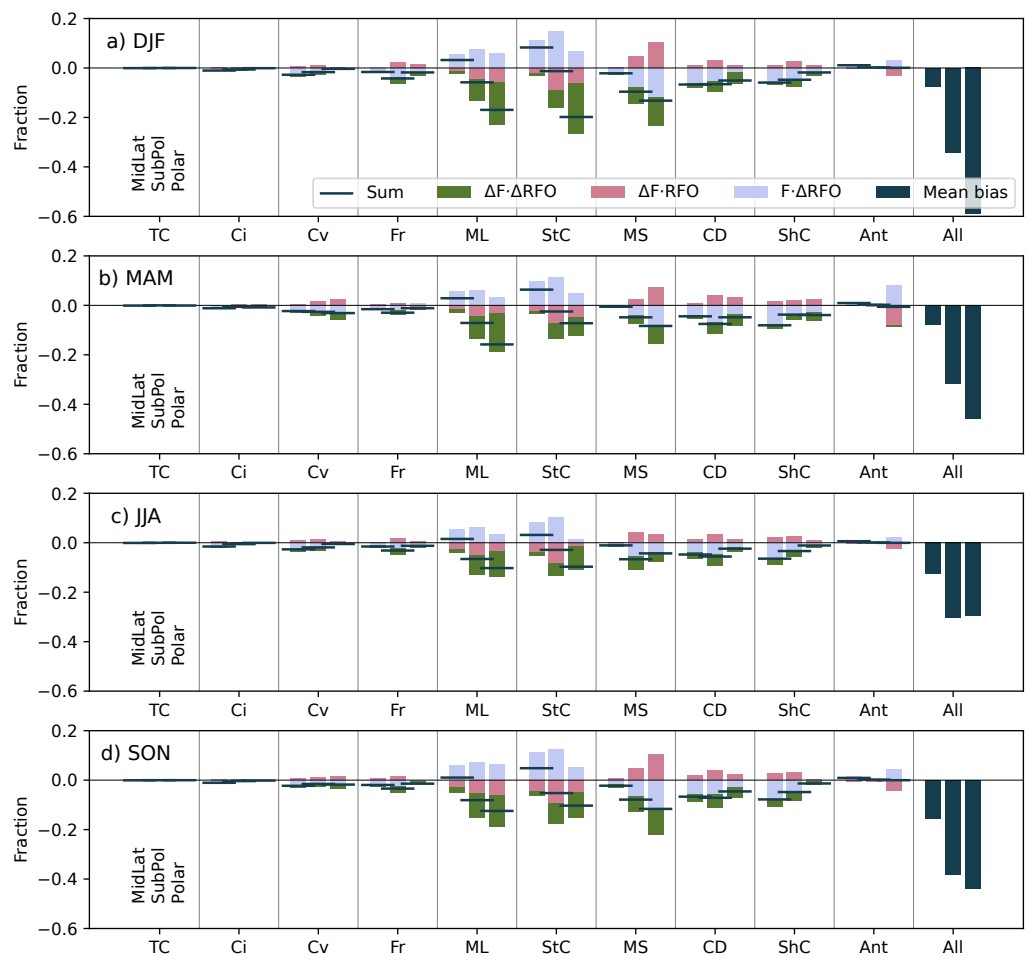

**Figure A1.** The decomposed mean biases in the CFL (fraction) for each cloud regime (thin cirrus, TC; cirrus, Ci; convective, Cv; frontal, Fr; mid-level, ML; stratocumulus, StC; marine stratiform, MS; cloud decks, CD; shallow cumulus ShC; Antarctic, Ant) over the three regions from left to right mid-latitudes, sub-polar and polar, for each season (DJF, plot a; MAM, b; JJA, c; SON, d respectively). The sum of the decomposed biases are shown by the horizontal bar, while the the terms of the bias decomposition (see eq. 1) are shown in blue ($F_r^{sat} \cdot \Delta RFO_r$), pink ($\Delta F_r \cdot RFO_r^{sat}$) and green ($\Delta F_r \cdot \Delta RFO_r$). On the far right of each plot is the mean radiative bias for each region/season as a reference

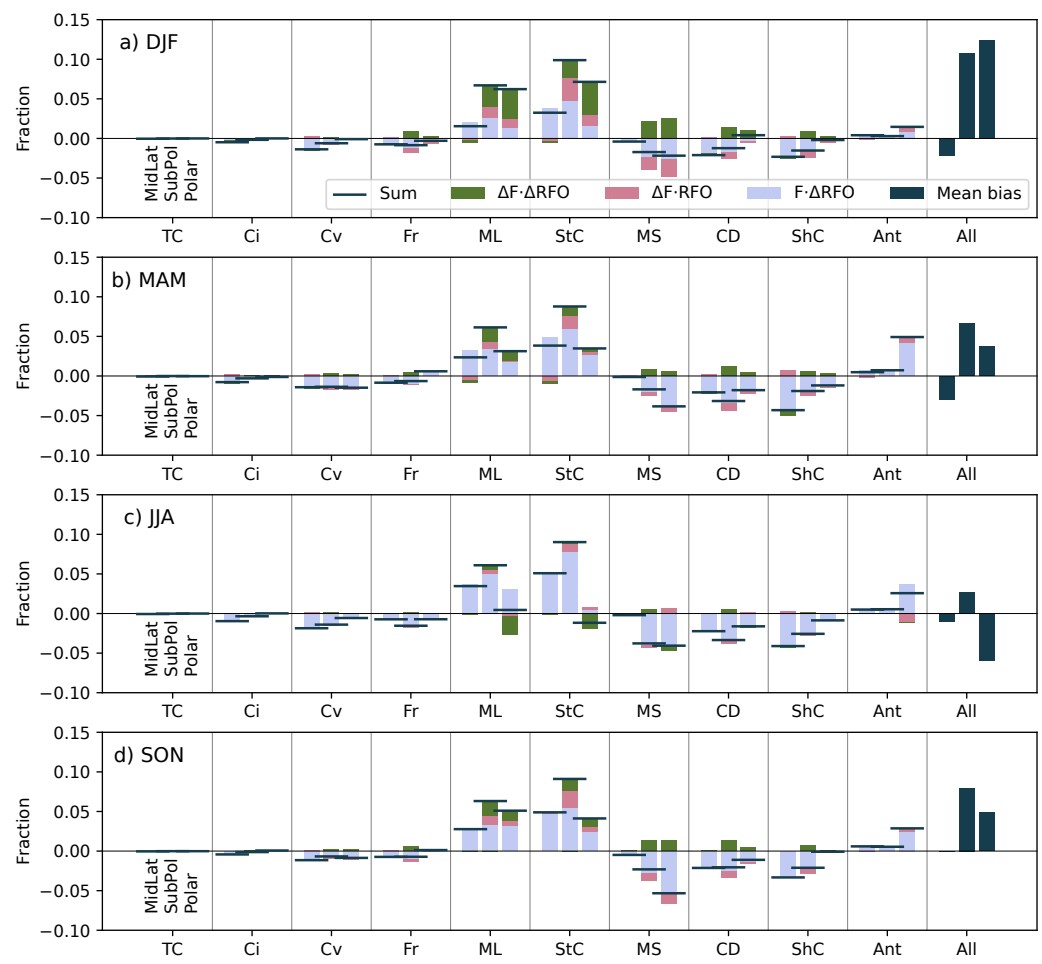

**Figure A2.** The decomposed mean biases in the CFI (fraction) for each cloud regime (thin cirrus, TC; cirrus, Ci; convective, Cv; frontal, Fr; mid-level, ML; stratocumulus, StC; marine stratiform, MS; cloud decks, CD; shallow cumulus ShC; Antarctic, Ant) over the three regions from left to right mid-latitudes, sub-polar and polar, for each season (DJF, plot a; MAM, b; JJA, c; SON, d respectively). The sum of the decomposed biases are shown by the horizontal bar, while the the terms of the bias decomposition (see eq. 1) are shown in blue ($F_r^{sat} \cdot \Delta RFO_r$), pink ($\Delta F_r \cdot RFO_r^{sat}$) and green ($\Delta F_r \cdot \Delta RFO_r$). On the far right of each plot is the mean radiative bias for each region/season as a reference

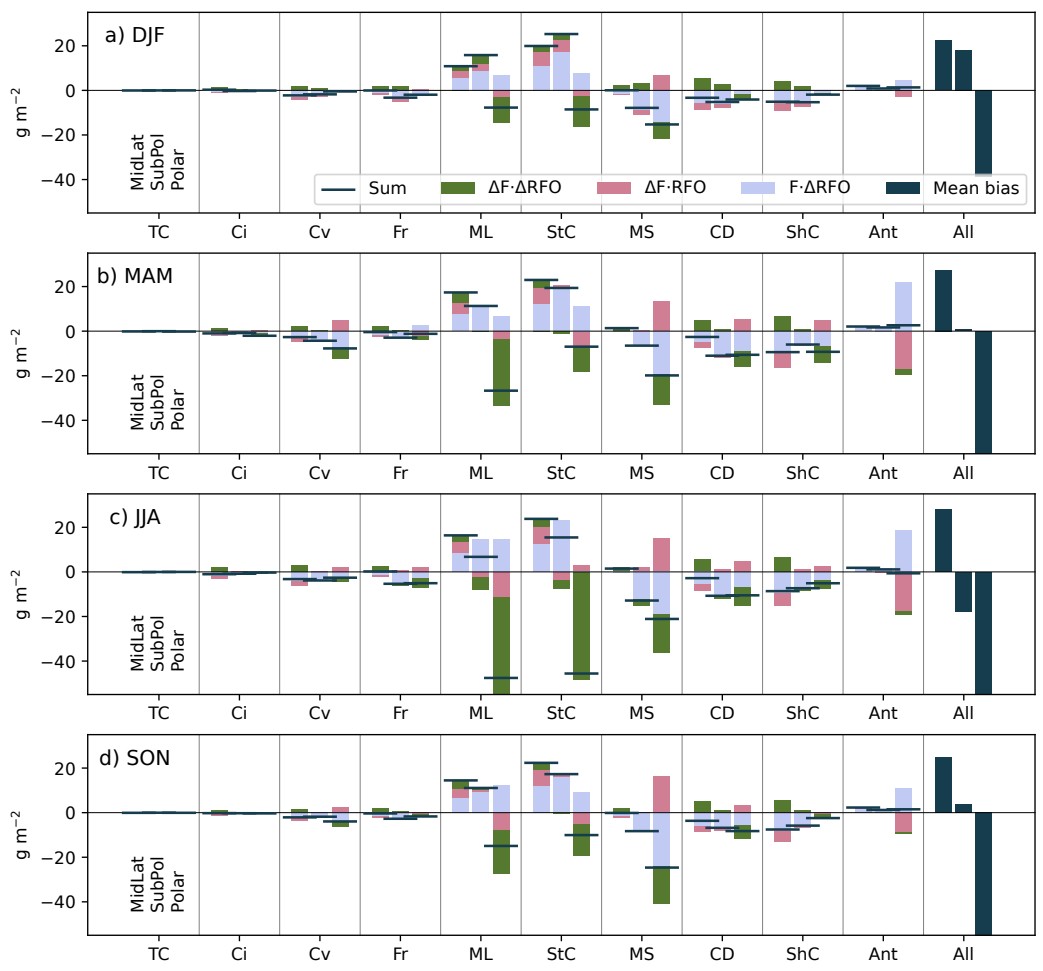

**Figure A3.** The decomposed mean biases in the LWP (g m$^{-2}$) for each cloud regime (thin cirrus, TC; cirrus, Ci; convective, Cv; frontal, Fr; mid-level, ML; stratocumulus, StC; marine stratiform, MS; cloud decks, CD; shallow cumulus ShC; Antarctic, Ant) over the three regions from left to right mid-latitudes, sub-polar and polar, for each season (DJF, plot a; MAM, b; JJA, c; SON, d respectively). The sum of the decomposed biases are shown by the horizontal bar, while the the terms of the bias decomposition (see eq. 1) are shown in blue ($F_r^{sat} \cdot \Delta RFO_r$), pink ($\Delta F_r \cdot RFO_r^{sat}$) and green ($\Delta F_r \cdot \Delta RFO_r$). On the far right of each plot is the mean radiative bias for each region/season as a reference

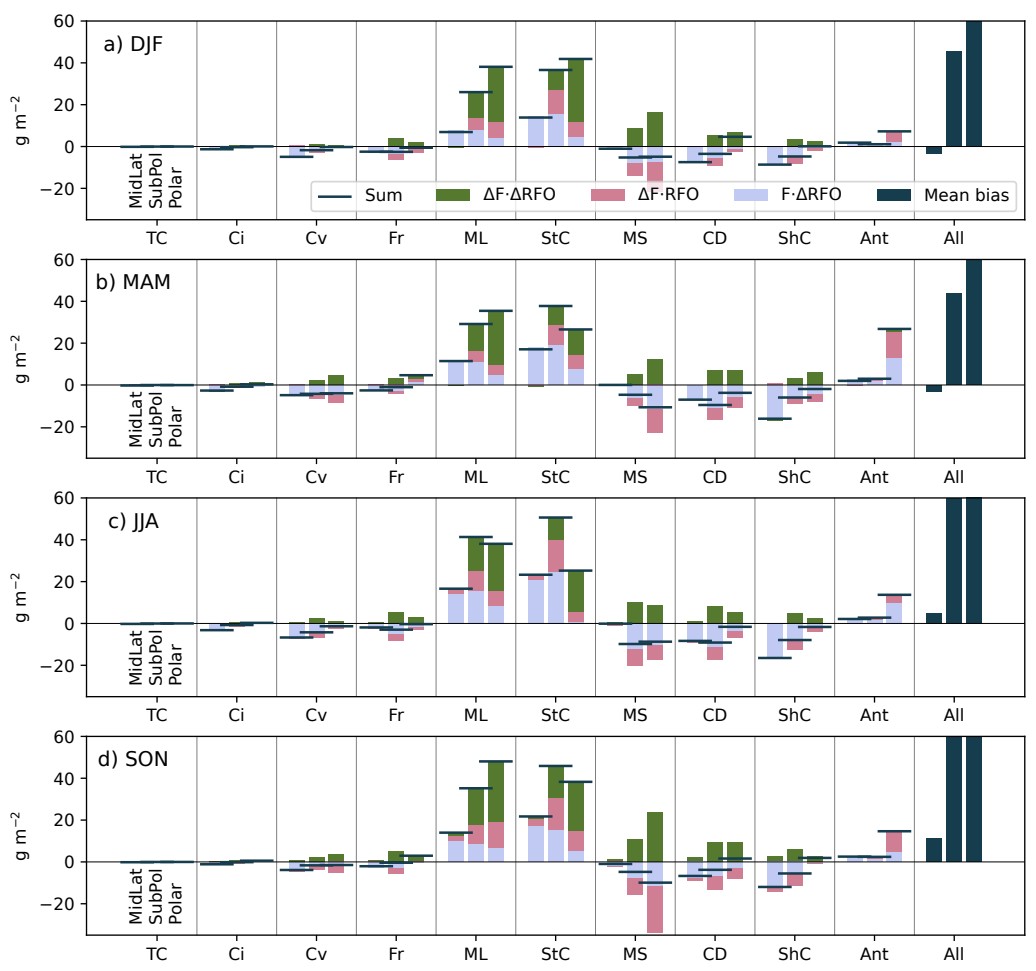

**Figure A4.** The decomposed mean biases in the IWP ($\mathrm{g\,m^{-2}}$) for each cloud regime (thin cirrus, TC; cirrus, Ci; convective, Cv; frontal, Fr; mid-level, ML; stratocumulus, StC; marine stratiform, MS; cloud decks, CD; shallow cumulus ShC; Antarctic, Ant) over the three regions from left to right mid-latitudes, sub-polar and polar, for each season (DJF, plot a; MAM, b; JJA, c; SON, d respectively). The sum of the decomposed biases are shown by the horizontal bar, while the the terms of the bias decomposition (see eq. 1) are shown in blue ($F_r^{sat} \cdot \Delta RFO_r$), pink ($\Delta F_r \cdot RFO_r^{sat}$) and green ($\Delta F_r \cdot \Delta RFO_r$). On the far right of each plot is the mean radiative bias for each region/season as a reference

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
