# Peer review of "Southern Ocean cloud and shortwave radiation biases in a nudged climate model simulation: does the model ever get it right?"

_Atmospheric Chemistry and Physics, 2022_

## Author Response (AR1)

**Author Response and Manuscript Revision**

Sonya Fiddes

June 2022

Dear Dr. Burrows,

Please find our response to the Reviewer's comments and our revised manuscript 'Southern Ocean cloud and shortwave radiation biases in a nudged climate model simulation: does the model ever get it right?'. We are thankful for the Reviewer's timely and detailed comments.

Both reviewers provided feedback that there was still some ambiguity in the description of our methods. We have spent significant time making the text clearer in response to this, including adding a new Figure as suggested by Reviewer 1. We believe that this has made our work easier to understand and that our ideas and new perspective on this problem are now clearly communicated.

In addition, in our revisions we realised that we had gone against convention for calculating the top-of-atmosphere CRE values, meaning that our CREs were of opposite sign to that of the literature. While our interpretation of the CRE remains correct (eg. that clouds are not reflecting enough radiation back out to space), we have now updated the figures and text to reflect the conventional sign of the CRE.

Below is the comment by comment description of our changes and attached is a track-changed copy of the updated manuscript. We hope you find these changes satisfactory for publication.

Kind regards,
Dr. Sonya Fiddes

**Response to Reviewer 1**

Many thanks to this Reviewer for their careful comments. We appreciate the time they have spent highlighting where our methods and ideas were not clearly communicated. In particular, we thank the Reviewer for the suggestion of a diagram to help communicate our work. We have now resolved some of the questions this Reviewer had around our methods. Our detailed response can be found below.

**Overall comments**

*The study presents a k-means cluster analysis of the recent ACCESS model in the spirit of (Williams and Webb 2009) and (Haynes et al. 2011). The paper could do more to concretely link biases in cloud properties to biases in radiation- radiative biases are suggested to be related to biases in various cloud properties, but this appears to be by eye rather than quantitative. In several places the writing is difficult to follow and especially in the analysis section it is hard to follow whether cloud RFO or cloud properties are being referred to (in several cases clouds are referred to as being simulated correctly or incorrectly, but it is unclear what that means in the regime framework) and often the ability of the model to replicate these quantities is described in vague relative terms. It is also not clear if the authors are comparing in-cloud and area-averaged water paths.*

We have carefully read through the manuscript to make sure we are consistent with our terminologies (correct/incorrect definitions) and that we remove any potentially vague statements. Examples of this are specifically outlined in the remainder of the comments below.

*Abstract: In several places the authors refer to incorrectly or correctly simulating clouds. It is ambiguous what they mean by this. It seems to be only referring to phase and frequency (which I think it equivalent to cloud fraction). If this is the case, it would be good to clarify that we only care about phase and frequency in the abstract and not other things like optical depth and condensed water path (for instance).*

We have rephrased the abstract in two ways: the first to make it clear that we are not just referring to phase and frequency, and the second to clarify what we mean when we refer to incorrect and correct simulation of clouds. We hope that this better communicates our research findings and methods.

Line 2: 'The radiative bias, characterised by too much shortwave radiation reaching the surface, is attributed to the incorrect simulation of cloud properties, including frequency and phase. To identify cloud regimes important to the Southern Ocean, we use $k$-means cloud histogram clustering, applied to a satellite product and then fitted to nudged simulations of the latest generation ACCESS atmosphere model. We identify instances when the model correctly or incorrectly simulates the same cloud type as the satellite product for any point in time or space. We then evaluate the cloud and radiation biases in these instances. We find that when the ACCESS model correctly simulates the cloud type, cloud property and radiation biases of equivalent, or in some cases greater, magnitude remain compared to when cloud types are incorrectly simulated.'

*L44: what ensemble is being referred to?*

We have clarified that this study used an ensemble of selected CMIP6 models.

Line 45: 'Shallow-cumulus clouds are found to be consistently underestimated by a selection of CMIP6 models, ...'

*L60: What aspect of Bodas-Salcedo 2014 demonstrates a need for consistent evaluation techniques?*

This sentence was not intended to convey that Bodas-Salcedo et al. (2014) demonstrates a need for consistent evaluation, but rather that it demonstrates an instance where evaluation techniques have been used across studies, highlighting the effectiveness of consistent techniques. We removed this sentence.

*L62: it is unclear what the first two sentences of this paragraph are referring to. What are climate-scale runs? Why wouldn't the synoptic meteorology be the same? I think what the authors are getting at*

*is the difference between coupled and AMIP runs. However, there is not any guarantee that the synoptic state will be the same across AMIP runs and the authors just discussed Field and Wood 2007 above, which composites on synoptic state- making it immaterial whether low pressure centers and other synoptic features are occurring in the same place.*

We have revised the first sentence of this paragraph to make clear that we are referring to free-running, long (climate-scale) simulations. We have revised the second sentence to make clear that often with these runs only monthly output is available (suiting the 'climate scale') and that the synoptic state (regardless of the temporal resolution of the output) cannot be considered the same as that observed due to the free-running nature of the models. The intention of these types of simulations is for climate analysis, where long-term averages can be considered directly comparable to observations. We have further revised this section to be clear that some of the simulations often used for theses analyses are in the AMIP style, with prescribed SSTs/sea ice, but this however still does not allow the synoptic states to be directly comparable to observations. The reviewer correctly notes that previous studies have gotten around this issue by using synoptic typing, when high-enough temporal resolution has been available. We point out that the synoptic typing done however often limits the analysis to particular conditions (eg. the predominant focus has been on cyclones, end even narrower, the cold sector of such) and ignores possible compensating errors in other synoptic situations. While cyclones are extremely common in the SO and may be the lead contributor of the radiative bias, other synoptic situations such as high pressure systems and frontal systems are also of worthy of examination. In this study, we wish to examine the entire system, without pre-conceived ideas of model performance (eg. limiting to one specific condition).

Line 62 : 'The majority of the aforementioned cloud-regime studies have compared free-running simulations, such as those performed for the CMIP experiments, including the Atmospheric Model Intercomparison Project (AMIP) where sea surface temperatures and sea ice concentrations are prescribed. However, with free-running simulations, the depth of analysis is limited as the synoptic scale meteorology cannot be considered the same. Some studies have used synoptic compositing to alleviate this issue, where certain synoptic situations can be compared like-for-like, and location and timing is then considered irrelevant. However, these studies are often limited to one synoptic type, such as cyclone centers, ignoring a number of other synoptic situations relevant for the SO, as well as any compensating errors that may exist. Additionally, focusing on just one synoptic condition follows a pre-conceived idea of the error, which may or may not hold true for newer model generations.'

*L75: It is somewhat vague what the authors mean by 'incorrectly' or 'correctly' simulated... Is this just in terms of phase and frequency, or in terms of all characteristics in a more abstract way?*

We have revised this sentence for clarity:

Line 81: 'By using a nudged simulation, we are able to composite and evaluate days and locations when cloud regimes are correctly (ie. are the same as what was seen by the satellite) identified by the model, as well as instances when the model incorrectly simulates the cloud regime. We aim to answer the following question: if the model simulates the correct cloud structure for the time and place, is the radiative bias improved?'

*L175- It's a little ambiguous here whether the authors are referring to IWP and LWP averaged over the grid box, which is what the model outputs (aka clivi and clwvi-clivi), or if they are talking about in-cloud liquid and ice water path, which is what MODIS would see. It is also somewhat mysterious how propagating errors would affect LWP and IWP and not other cloud properties in COSP. Some additional discussion of this is needed to instill confidence in their evaluation.*

We have used the grid box mean and performed the appropriate calculations to ensure the fields are comparable. We have made this clear in the text.

Line 186: 'For the LWP and IWP, we are considering the grid box mean (as opposed to the in-cloud mean).'

We have also expanded our discussion of propagating errors. Here we explicitly refer to the documented poorer retrieval of the cloud effective radii, which is used in the calculation of the LWP/IWP,

hence the errors of this top-level retrieval are passed through to the LWP/IWP. We have chosen to use the raw model fields in this instance because we do not trust the retrievals of effective radii (which we also evaluated, but have not shown), and their impact on LWP/IWPs. We have added the text below to clarify this point and removed reference to propagating errors.

Line 187: 'IWP and LWP are reliant on $R_{eff}$ retrievals, which as discussed above, are less well captured in satellite products.'

*L193: Consistent with which previous studies?*

We have removed this statement.

*L235 and 245: Is CFL/CFI random overlap, or just what is seen from space? Could biases be driven mostly by this cirrus in the model if it is just what is seen from space, with minimal relevance for the PBL cloud that drives SWCRE?*

As we understand, the CFI/CFL fields are the fraction of pixels successfully retrieved by MODIS, as seen from space, which the ACCESS-AM2 model replicates with the COSP simulator package. The point you make about cirrus is a good one, however, in the Southern Ocean, these cloud types are very infrequent (eg. see Mace et al. 2009 https://doi.org/10.1029/2007JD009755) so we do not expect them to be playing a large role in the SWCRE.

*L249: Consider citing: Mülmenstädt, J., Salzmann, M., Kay, J. E., Zelinka, M. D., Ma, P.-L., Nam, C., et al. (2021). An underestimated negative cloud feedback from cloud lifetime changes. Nature Climate Change, 11(6), 508–513.*

*Field, P. R., & Heymsfield, A. J. (2015). Importance of snow to global precipitation. Geophysical Research Letters, 42(21), 9512–9520.*

We thank the reviewer for bringing these papers to our attention. We have now removed reference to precipitation in this particular sentence, but we have included the Mulmenstadt work in our discussions.

Line 265: 'Too few liquid clouds which are instead simulated as ice clouds, will result in clouds that are more optically thin causing not enough short wave radiation to be reflected out to space.'

Line 611: 'Furthermore, in this work we have only considered a few cloud properties that impact the radiation budget. Other important factors, such as the number and size of cloud droplets, precipitation phase and amount, or other thermodynamical properties are likely to impact the absorption and scattering properties of clouds and their lifetime (for example: Mulmenstadt et al. 2021).'

*L253: again, it is unclear if the authors are comparing in-cloud LWP and IWP to area- mean LWP and IWP.*

Please see our response to the previous comment on the definition of LWP and IWP.

*L273: This discussion is fairly qualitative in terms of relating various cloud properties to radiative bias. Quantitative estimates of how (for instance) cloud fraction relates to radiation exist:*
*Bender, F. A. M., Engström, A., Wood, R., & Charlson, R. J. (2017). Evaluation of Hemispheric Asymmetries in Marine Cloud Radiative Properties. Journal of Climate, 30(11), 4131–4147. https://doi.org/10.1175/JCLI-D-16-0263.1*
*Can the authors show whether the CF bias in their simulations can explain the actual radiative bias?*

We thank the reviewer for bringing this paper to our attention. We have added this reference to our conclusion section.

We agree that this paper is more qualitative in its analysis, though we do not believe this work is an exception to the literature norms in that respect. We currently have in preparation a follow up paper that takes a much more quantitative view on diagnosing the relationship between cloud properties (including cloud fraction) and radiative biases. We will submit this work shortly (pending this paper's acceptance).

Line 602: 'The work presented in this study has provided a relatively qualitative view of the cloud and radiative biases associated with cloud types. Studies such as Bender et al. (2017) have used more quantitative methods to evaluate the role cloud fractions play in determining the radiative balance. Specifically, Bender et al. (2017) examine the distributions of cloud albedo and the associated cloud fraction in CMIP5 models and their linear relationships. In further exploration of the data used in this work, we note that most of the relationships between cloud properties and radiative bias are in fact non-linear, which is also highlighted in this work by the differing relationships observed by latitude/cloud type. Similar results have been noted in Bodas-Salceado et al. (2016), where the LWP and radiative biases were not found to be as tightly coupled as expected. We will present findings of a quantitative analysis employing machine learning to understand the role cloud properties play in determining the radiative bias in an upcoming paper.':

*Section 5.2.1: this section and the associated figure 7 are quite hard to follow. The authors may benefit from more clearly distinguishing errors in RFO and in cloud properties for a given cluster. The writing is somewhat ambiguous- clouds are referred to as being 'correctly' simulated- is this in terms of getting enough of them, or in terms of them looking right when they show up? In particular, the second paragraph of this section could be improved by using fewer vague qualifiers ('comparatively well captured', 'relatively well captured', 'somewhat correct',...) what is the baseline for these statements? These statements are then used to make causal statements about what biases in clouds are leading to biases in radiation, but without any support – wouldn't it be possible to do a more quantitative assessment of where biases in SWCRE are coming from?*

We have revised the beginning of Section 5 to better describe our terms 'correct' and 'incorrect' and we have used this language more consistently throughout the text.

Line 389: 'One strength of comparing a daily, nudged, simulation to daily MODIS fields is the ability to make direct comparisons in time and space. As the synoptic meteorology is considered to be the same in the model and the observed conditions, we therefore expect that the model microphysics, if accurate, would generate the same cloud types that the large scale dynamics prescribes. With this assumption, we are able to isolate instances (points in time and space) where the model simulates the same cloud type as MODIS, which we define as 'correctly' simulating the cloud type, and the instances where the model simulates a different cloud type, which we define as an 'incorrect' cloud type assignment by the model. We demonstrate these definitions in Figure 6a and b. '

*Figure 7 is pretty hard to follow. The authors may need a cartoon with annotations or something to illustrate this. There are dots, colors, outlines, months, clusters, and 5 different quantities. A single cartoon for one of the subplots would be helpful.*

We have now included a diagram that helps to explain Figures 6, 7, 8 and 9. We have also tried to reduce the complexity of these figures by removing some of the information.

*Overall, I would suggest moving the summary of the findings before section 5.2.1 to give the reader an overview of what is going to be discussed.*

We have significantly revised Section 5 to improve clarity. We have also revised the introducing paragraphs so that the reader has a clearer idea of what to expect. For these reasons, we have not moved the summary to the top.

*L555: the authors bring up a good point- is some of their RFO bias simply due to nudging? Can the k-means clustering be replicated on a free-running simulation to see what the biases look like?*

A study examining the effect of nudging on cloud properties would be of interest, however is not within scope for our funding. We note that for cloud studies, nudging the temperature can change homogeneous ice nucleation rates, impacting cloud phase (Zhang et al. 2014, doi:10.5194/acp-14-8631-2014). This may influence other cloud properties, such as the optical depths and heights used in this work to develop the cloud types and subsequently the RFOs. However, Zhang et al. (2014) also showed that the largest impacts of nudging were felt in the tropics, including particular for convection. We speculate that due to the predominantly large-scale nature of the Southern Ocean (eg. fewer convectively driven clouds) and with temperatures already close to zero or below, the impacts of nudging will be less.

**Response to Reviewer 2**

We thank Reviewer 2, Alex, for your positive comments and constructive feedback on our manuscript. We have expanded on our methods to make more clear how and why we have chose the methods we did and we have tried to include more discussion of SCLW throughout. We have now included one of the Figures requested in the main text, while the others are in supplementary material. We have addressed each comment below.

**Major comments**

*In the current manuscript the normalisation of the cloud top pressure - cloud optical thickness joint histograms is ambiguous. For the majority of prior papers using these histograms, they are normalised by the cloud fraction value so that the sum of the cells of the histograms add to the cloud fraction value. If I had to guess based on figure 2, each of the histograms in this study are normalised to a value of 1 (I could be wrong about this). If this is the case it will have some implications on the interpretation of the results with respect to prior studies. This alternative normalisation could be justified by arguing the paper's focus is on phase and vertical structure which may be better captured with this approach, however currently I cannot find any discussion of this in the paper and it should definitely be discussed. While not expected in this paper, it could be interesting to compare results of these different normalisations.*

Thank you for raising this issue. Yes you are correct, we have normalised our histograms to one, and not to the cloud fraction. We have found that this has yielded better results for our clustering methods, as it reduces the effect of biases in cloud fraction on the clustering. ACCESS-AM2 has significant cloud fraction biases, which is shown by the ice and water phase cloud fractions discussed within, but is also true for the total cloud fraction which has not been discussed in this work. If we normalise the histograms to the cloud fraction, we get sensible cloud regimes from the MODIS product, as expected, but when this is applied to the ACCESS-AM2 histograms, the majority of points are assigned to a singular low cloud fraction cluster (eg. less than 30% coverage), with no real height or optical property definition. This considerably effects our analysis. By normalising to one, the clustering focuses more on the clouds vertical profile/optical depths, without the biases introduced by the ACCESS-AM2s poor simulation of cloud fraction. This is not to say that the cloud fraction bias is unimportant, but we discuss this bias in detail in other parts of the manuscript. We have attached the figures for both methods here for your interest (Figures 1 and 2) and we have also made sure to clarify our methods in the text.

Line 197: 'We have normalised the CTP/$\tau$ histograms to one (as opposed to the cloud fraction) to limit the impact of cloud fraction biases within the ACCESS-AM2 model on the clustering results. Whilst this impacts our ability to compare to other studies, it allows the clustering to target cloud vertical extent and thickness regardless of the total fraction and how well it is captured by the ACCESS-AM2 model. '

*I am surprised by the lack of discussion of supercooled liquid water throughout the manuscript. There are several places, particularly in your results, where I think some discussion is warranted. Many of your results show too much ice fraction and not enough liquid in key cloud types which is indicative to me of issues with the model representation of supercooled liquid water. Some good places to add this would be the paragraph starting on line 273, the discussion of figure 7-9 and the conclusions.*

Thank-you for pointing this out. We have now including discussion of SCLW throughout the manuscript as appropriate, including below:
Line 286: 'This finding agrees well with the literature in that not enough liquid water exists below zero degrees Celsius, instead being simulated as ice (Bodas-Salcedo et al. 2016)'

Line 294: 'The role of cloud phase, and in particular, that of supercooled liquid water, appears to have significant latitudinal dependence, likely influenced by a range of factors, including temperatures and ice nucleating particle (INP) availability.'

*There are some figures that are passingly discussed in the text but not currently in the paper which would make great additions to an appendix. In particular, I am thinking about the phase property versions of Figure 5 and the individual sub-region versions of Figure 6. I know I would be interested in seeing*

[Figure]

Figure 1: Clustering when normalised to one, as done in this study

*those figures and the most appropriate place via ACP guidelines appears to be in an appendix.*

We have added the cloud phase versions of Figure 5 as supplementary material. We have now replaced Figure 6 with the three subplots of the sub-regions.

**Minor comments**

*Line 7: Sentence starting on this line should be simplified due to its complex clausal structure.*

We have altered this sentence as follows:

Line 9: 'We find that when the ACCESS model correctly simulates the cloud type, cloud property and radiation biases of equivalent, or in some cases greater, magnitude remain when compared to when cloud types are incorrectly simulated.'

*Line 20: Consider changing "simulation by models of cloud properties" to "simulation of cloud properties within models"*

Sentence changed as suggested.

*Line 46: Consider changing "compensate the" to "compensate for the"*

Sentence changed as suggested.

*Line 84: I think something has gone wrong with the citation formatting here*

Yes, citet instead of citep. Fixed now.

[Figure]

Figure 2: Clustering when normalised to the cloud fraction

*Line 153: Can you please be more specific about the identification of clear sky cases and their removal from the dataset*

Instances of clear sky were determined by summing the histograms. Any histogram that added up to zero was removed from the data set with which clustering was applied to.

Line 162: 'We have done this by finding the instances where the CTP-$\tau$ histograms summed to zero. '

*Line 190: I think you have cited the wrong paper by mistake here. From what I can see Pendregosa 2011 does not have that information.*

We were referring to the SciKit Learn API, which Pedregosa 2011 is the reference for. We have made this more clear.

Line 205 'The SciKit Learn application programming interface provides a detailed explanation of each metric in addition to their advantages and disadvantages.'

*Section 3 could possibly do with some references to past paper which have identified similar seasonal (Bodas-Salcedo et al. 2012) or spatial (eg. Kuma et al. 2020) biases*

We have included references to these papers in this section as suggested.

Line 234 'The summer (DJF), shown in Figure **??**c1, continues to have the largest polar bias, while the winter (JJA) season has the smallest bias overall, in agreement with previous work (Bodas-Salcedo et al. 2012, Kuma et al. 2020).'

Line 244 'Kuma et al. (2020) also noted a strong latitudinal dependence of radiative biases in a similar version of the UM and a reanalysis product.'

*Line 225: Consider changing "less zonal" to "less zonally coherent"*

Sentence changed as suggested.

*Figure 1 appears to have an issue where some rows are more magnified than others leading to some straight edges*

We have amended the Figure to resolve this issue.

*Line 261: I think "indicating that this is a complex system to understand" should be changed to something that stresses the behaviour in the system is complicated inplace of how understandable it is.*

I have removed 'to understand' to this effect.

*Line 282: Simplify the wording of "using 12 clusters for five years of daily-mean joint histograms"*

Sentence simplified to:

Line 304: 'The $k$-means clustering technique, spanning five years of daily-mean joint histograms over the entire globe, resulted in the 12 cluster centres shown in Figure 2.'

*Line 288: Consider deleting "while this is important to note,"*

Deleted.

*Figure 3 and 4: The last sentence in the caption is a little unclear and awkwardly worded. Consider simplifying.*

Simplified to:

Figure 3 & 4 captions: 'The numbers in each title represent the mean frequency of occurrence for each cloud regime (in time and space) over SO region'

*Line 327: Consider deleting "region of interest"*

Deleted

*Line 328: Consider rewording "are spatially consistent in sign and for some, magnitude" as it is a little confusing.*

We have removed 'and for some, magnitude' to this effect

*Line 333: The sentence stating on this line may need some rewording as it could imply you do not look at subregions instead of the intended meaning that non-SO data is excluded.*

We have amended this sentence:

Line 355: 'Note that from this point on we only consider the broad SO region, and the three sub-regions defined within.'

*Line 371: I found the wording of the first three sentences of this paragraph quite hard to follow. I would suggest rewriting them with an emphasis on clarity.*

We have significantly revised this section to outline more clearly our methods. We have also added another figure to help aid the readers understanding (Figure 6).

Line 403: 'Each panel of Figure 7 shows for each MODIS cloud type (y-axis), the percent of time that each cloud type is assigned by ACCESS-AM2 (x-axis). The total number of instances that that cloud type is observed by MODIS for each region is shown on the right. If ACCESS-AM2 simulated every cloud type the same as what the MODIS product did, we would expect a diagonal line through Figure 7 of 100%, with zeros elsewhere. '

*Section 5.2.1 even though it is incredibly rare, a sentence discussing the extreme phase based biases shown in the TC class could be a valuable addition here.*

We have added a sentence discussing the TC biases. We do not feel that we can draw much for this results, as this type of cloud is occurring approximately 85 times in the the whole data series (see Figure 6, 8547 / 1% ).

Line 474: 'The thin cirrus clouds are shown to have large biases in the spring (SON) when correctly simulated for all fields. However, we note that these cloud types occur infrequently (see Figure ??a, total number for thin cirrus clouds) and are only correctly simulated 1% of the time, and hence, we do draw any conclusions from this result. '

*Line 401: The transition to a new sentence discussing mid-level clouds feels off, because I assume the previous sentences were already discussing them. I think this could probably just be resolve with some more clear wording*

We have significantly revised this paragraph (and section) to make it clearer.

Line 428: 'For the mid-level and marine stratiform clouds, the $SWCRE_{TOA}$ biases are larger when the cloud types are assigned incorrectly (outlined bars) than when they have been correctly simulated by ACCESS-AM2 (coloured bars). However, when these cloud types are correctly assigned (which happens 63% and 12% of the time respectively), the $SWCRE_{TOA}$ bias are for most cases non-negligible in most seasons. For the mid-level clouds, the $SWCRE_{TOA}$ biases are smaller in magnitude when the clusters are correctly identified by the model. Interestingly, this is not the case for the CFL and LWPs, which both have larger biases when the clusters agree, suggesting that the radiative effects associated with too few liquid water clouds is partially compensated by them being too optically thick. This indicates that the lower $SWCRE_{TOA}$ bias when the mid-level clouds are correctly simulated may be occurring for the wrong reasons. For the marine stratiform clouds, the CFL is strongly underestimated when it is incorrectly simulated, which is expected to produce negative $SWCRE_{TOA}$ biases. The CFI is comparatively well simulated. Interestingly, the LWP and IWPs seem to be relatively well captured for marine stratiform clouds. This suggests that the $SWCRE_{TOA}$ bias may be predominantly driven by an underestimated CFL, while the amount of water in them is somewhat correct.'

*For figures 7-9 I think the exact definition of the total column is unclear. I can't determine if it shows the overall biases associated with accurately simulated clouds or with all clouds. If it is showing for both "incorrectly" and "correctly" represented cloud types combined, it could be interesting if there was a way to decompose this information and show each independently. Regardless of if this is possible or not, some text needs to be added clarifying what this column represents.*

We have removed the total column here to avoid confusion.

*Line 446: The sentence starting on this line is confusing due to its clausal structure. Can you simplify the sentence or split it into two.*

We have rephrased as follows:

Line 482: 'The mid-level and stratocumulus clouds at the two most accurately simulated cloud types (68% and 66% of the time). The cloud radiative biases in these cases are relatively low, implying that the model may be doing a good job.'

*Line 449: Consider deleting the "For" at the start of the sentence.*

Removed.

*Line 454: Please simplify "Considering now what may be contributing to"*

We have rephrased as follows:

Line 491: 'We can see that the majority of the positive bias in the sub-polar region is driven by shallow cumulus clouds, that are incorrectly simulated.'

*Line 531: I think the sentence starting on this line is a little confusing and would benefit from being rewritten.*

We have simplified this sentence as follows:

Line 571: 'Marine-stratiform, stratocumulus and mid-level clouds contribute most to the polar region's negative $SWCRE_{TOA}$ bias.'

*Line 533: Consider deleting "biases, again whether the cloud regime is correctly simulated or assigned as something else,"*

We have revised as follows:

Line 571: 'Each of these cloud types are associated with large biases in cloud properties, including too few liquid clouds, too little LWP and too much IWP. This is the case both for when these cloud types are correctly and incorrectly assigned.'

---

## Author Response (AR2)

**Author Response and Manuscript Revision**

**Sonya Fiddes**

**September 2022**

Dear Dr. Burrows,

Please find our response to the second round of Reviewers comments and our revised manuscript 'Southern Ocean cloud and shortwave radiation biases in a nudged climate model simulation: does the model ever get it right?'.

We have addressed all of Reviewer 2's comments, and have tried to more carefully explain our reasoning with respect to Reviewer 1's comments. We appreciated Reviewer 1's concerns but are confident that what we have done is not incorrect. We hope that our further explanation and changes to the text satisfies both you and the Reviewer.

Kind regards,

Dr. Sonya Fiddes

**Response to Reviewer 1**

We thank this reviewer for taking the time to consider our revisions. We have addressed the comments below and hope that our answers address their concerns. We believe that our method is robust, though we acknowledge that a different choice may lead to different results. Please see our detailed comments below.

*There remains one serious issue, which is that the authors appear to be comparing clivi and clwvi-clivi (see line 195) to MODIS in-cloud IWP and LWP. It is stated: "For this reason, as stated above, the actual simulated (i.e. direct model output) IWP and LWP are used for this analysis instead of the COSP product. "*

We would like to reassure the reviewer that we are not comparing two different products in our analysis. We have carefully reviewed each product used in this work to ensure we are comparing the correct version (in-cloud versus grid box mean) and units. As part of this process, we have consulted with both UM and MODIS experts to seek clarification on aspects of the data sets including units and confirmation of grid-box vs in-cloud mean output.

Firstly, we are comparing clivi and clwvi (not clwvi-clivi as suggested) separately. The sentence stated above refers to our choice to use the raw model output as opposed to the model output that has been run through the COSP simulator. We will touch on this more below and have slightly modified the aforementioned sentence:

Line 196: 'For this reason, as stated above, the actual simulated (i.e. raw model output) IWP and LWP are used for this analysis instead of the COSP derived product.'

Both the COSP and raw modelled LWP and IWP are output as grid-box mean from the ACCESS model. The MODIS data, as the reviewer states, is an in-cloud mean estimate. So in order to satisfactorily compare the two, a conversion is needed, where:

```
grid box mean = in-cloud mean x cloud fraction
```

We have compared both methods. Converting the MODIS product to grid box mean as well as the model products to in-cloud mean (see here) reveals that there is no completely satisfactory solution. Comparing in-cloud values lead to very large model biases, where as the grid-box mean values were better represented for the LWP, but not the IWP. This is likely due to their weighting by cloud fraction. We have now made this clear in the text.

Line 162: 'Furthermore, care has been taken to ensure that we are comparing only grid-box mean values of the LWP and IWP. Model outputs (both COSP and raw) are provided directly as grid-box mean, while the MODIS products are provided as in-cloud mean values. We have performed the appropriate conversions where grid-box mean is equal to the in-cloud mean multiplied by cloud fraction. After comparison of both grid-box mean and in-cloud mean values, we have chosen to use the grid-box mean values.'

Similarly, as mentioned above, we have compared the COSP model output as well as the raw model output. The magnitudes of the raw model output were marginally better when compared to MODIS than that of the COSP model output, leading to our decision to use the raw output in this instance.

*If you are using these outputs there isn't really an observational IWP for grid-box mean, but you can use microwave radiometer output.*

We appreciate that the converted IWP grid box mean may not be the best product to use, however we do not wish to mix and match products. The advantage of the MODIS products is that we can use the retrievals for a number of cloud properties to directly compare with our model in a consistent manner. Although other products and measurements could be used to evaluate each cloud property separately, that would need to be a different study that would carefully deal with the caveats of the inconsistencies in measuring or retrieving the different cloud products to compare to a model.

*Swapping in-cloud and area-mean LWP between GCM output and observations would seem to explain Fig1 a4 and b4 where there is a big latitudinal bias that seems like it follows the distribution of cloud*

*fraction since in-cloud LWP\*CF  area-mean LWP.*

Yes, the reviewer is right that there is some dependence of the LWP (and IWP) on the cloud fractions, as we have used the grid-box means. We have made further note of this in the text:

Line 166: 'This choice does effect some of the results of this study, as would choosing to use the in-cloud values instead. However, we believe this choice is robust for two reasons: firstly, grid-box means are the native model output and this is a model evaluation study and secondly, the grid-box means showed a better model performance than the in-cloud mean values, likely due to the weighting of the cloud fraction.'

*It should also be noted that in the caption for figure 1 the units of a path are g/m2 not g/m3.*

We have fixed the units in this caption

*Because it is hard to say what swapping out the ice and liquid path variables will do I mark this major revisions.*

We appreciated the Reviewer's concern with this aspect of our methodology and we hope that we have been able to convince them that what we have done is robust, within the limits of the data available. We have included further discussion about these choices in our conclusion to ensure that we are acknowledging the caveats raised by this Reviewer.

Line 600: 'Similarly, having to convert between in-cloud and grid-box mean water paths has added another layer of uncertainty to this work. We chose to use the grid-box mean values, but the use of in-cloud values would have changed the our results and their interpretation. Whilst the COSP framework and specifically derived MODIS product go a long way to helping model evaluation, more needs to be done to ensure that modellers can use an observational products with confidence and minimal transformation. '

**Response to Reviewer 2**

We have addressed all of the minor comments from Reviewer 2 below.

*Sentence starting on line 180 has a slightly confusing clausal structure maybe "The MODIS products provide the best coverage over the time period examine, hence we choose to focus on MODIS data"*

We have revised the sentence as follows:

Line 188: 'The MODIS products provided the best coverage for the time period of interest to this study. For this reason, no other satellite products are considered in this work.'

*I think the sentence starting on line 183 needs to be rewritten for clarity*

We have revised the sentence as follows (and also removed the previous one).

Line 191: 'Using COSP output allows the appropriate comparison of model to satellite products. This method applies the assumptions and limitations of the satellite algorithms to the model output, limiting the possibility that biases are due to differences in processing.'

*The sentence starting on line 215 should be simplified. I think something like "This choice will allow us to assess if any changes applied to the model focussed on improving Southern Ocean clouds have unintended effects outside this region."*

We have amended this sentence following the reviewers recommendation.

Line 223: 'This choice will allow us to assess if any changes applied to the model focused on improving Southern Ocean clouds have unintended effects outside this region.'

*Line 227 change represent to "are set so that" and then change rest of sentence to fit*

We have revised the sentence to:

Line 235: 'The boundaries of our analysis, shown by the dashed lines, represent three regions of interest: the mid-latitudes defined as 30-43°S, the sub-polar region defined as 43-58°S, and the polar region defined as 58-69°S of the SO.'

*I think the sentence starting on line 263 needs to be rewritten for clarity*

We have revised the sentence to:

Line 271: 'In the mid-latitude region however, both the cloud fraction biases (liquid and ice) are weakly negative. This indicates too few clouds overall in this region, which can explain the positive $SWCRE_{TOA}$ bias.'

*I think the sentence starting on line 293 needs to be rewritten for clarity*

We have revised the sentence to:

Line 301: 'These results show that in the polar region, the biases in cloud fraction and water paths can satisfactorily explain the $SWCRE_{TOA}$ bias. For the other two regions however, the influence of the cloud biases is not as clear cut.'

*I don't think the sentence starting on line 304 makes sense. Maybe change the start to "The k-means clustering technique was applied to MODIS daily joint histograms...."*

We have amended the sentence to:

Line 312: 'The $k$-means clustering technique was applied to five years of MODIS daily-mean joint histograms over the entire globe. The 12 resulting cluster centres are shown in Figure 2.'

*I think the sentence starting on line 390 needs to be rewritten for clarity as it has a complex clausal structure.*

We have split this sentence into two as follows:

Line 388: 'The synoptic meteorology is considered to be the same in the model and the observed conditions, due to the nudging of the model. We therefore expect that the model microphysics, if accurate, would generate the same cloud types that the large-scale dynamics prescribes.'

*I think the wrong citation command is used for the bender et al. 2017 reference on line 604*

Fixed